# ACCELERATING DIFFUSION TRANSFORMERS WITH TOKEN-WISE FEATURE CACHING

**Chang Zou**[1,2*]  **Xuyang Liu**[3*]   **Ting Liu**[4]   **Siteng Huang**[5]   **Linfeng Zhang**[1†]
[1]Shanghai Jiao Tong University    [2]University of Electronic Science & Technology of China
[3]Sichuan University    [4]National University of Defense Technology    [5]Zhejiang University
**Code:** https://github.com/Shenyi-Z/ToCa

## ABSTRACT

Diffusion transformers have shown significant effectiveness in both image and video synthesis at the expense of huge computation costs. To address this problem, feature caching methods have been introduced to accelerate diffusion transformers by caching the features in previous timesteps and reusing them in the following timesteps. However, previous caching methods ignore that different tokens exhibit different sensitivities to feature caching, and feature caching on some tokens may lead to $10\times$ more destruction to the overall generation quality compared with other tokens. In this paper, we introduce token-wise feature caching, allowing us to adaptively select the most suitable tokens for caching, and further enable us to apply different caching ratios to neural layers in different types and depths. Extensive experiments on PixArt-$\alpha$, OpenSora, DiT and FLUX demonstrate our effectiveness in both image and video generation with no requirements for training. For instance, $2.36\times$ and $1.93\times$ acceleration are achieved on OpenSora and PixArt-$\alpha$ with almost no drop in generation quality.

## 1 INTRODUCTION

Diffusion models (DMs) have demonstrated impressive performance across a wide range of generative tasks such as image generation (Rombach et al., 2022) and video generation (Blattmann et al., 2023). Recently, the popularity of diffusion transformers further extends the boundary of visual generation by scaling up the parameters and computations (Peebles & Xie, 2023). However, a significant challenge for diffusion transformers lies in their high computational costs, leading to slow inference speeds, which hinder their practical application in real-time scenarios. To address this, a series of acceleration methods have been proposed, focusing on reducing the sampling steps (Song et al., 2021) and accelerating the denoising networks (Bolya & Hoffman, 2023; Fang et al., 2023).

Among these, cache-based methods (Ma et al., 2024b; Wimbauer et al., 2024), which accelerate the sampling process by reusing similar features across adjacent timesteps (*e.g.* reusing the features cached at timestep $t$ in timestep $t - 1$), have obtained abundant attention in the industrial community thanks to their plug-and-play property. As the pioneering works in this line, DeepCache (Ma et al., 2024b) and Block Caching (Wimbauer et al., 2024) were proposed to reuse the cached features in certain layers of U-Net-like diffusion models by leveraging the skip connections in the U-Net. However, the dependency on the U-Net architectures also makes them unsuitable for diffusion transformers, which have gradually become the most powerful models in visual generation. Most recently, FORA (Selvaraju et al., 2024) and $\Delta$-DiT (Chen et al., 2024b) have been proposed as a direct application of previous cache methods to diffusion transformers, though still not fully analyzed and exploited the property of the transformer-architecture. To tackle this challenge, this paper begins by studying how feature caching influences diffusion transformers at the token level.

**Difference in Temporal Redundancy:** Figure 1 shows the distribution of the feature distance between the adjacent timesteps for different tokens, where a higher value indicates that this token exhibits a lower similarity in the adjacent timesteps. It is observed that there exist some tokens that show relatively lower distance (in light blue) while some tokens show extremely higher distance (in dark blue), almost $2.5\times$ larger than the mean distance, indicating caching such tokens can lead to an overlarge negative influence. This observation indicates that *different tokens have different redundancy across the dimension of timesteps*, (*i.e.* different temporal redundancy).

---

*Equal contribution. shenyizou@outlook.com †Corresponding author: zhanglinfeng@sjtu.edu.cn

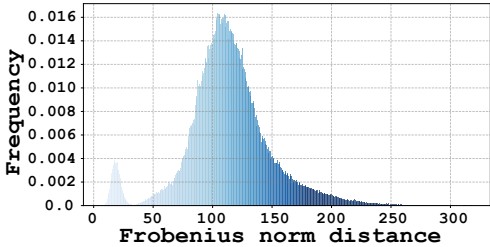
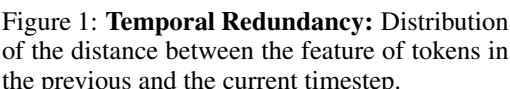
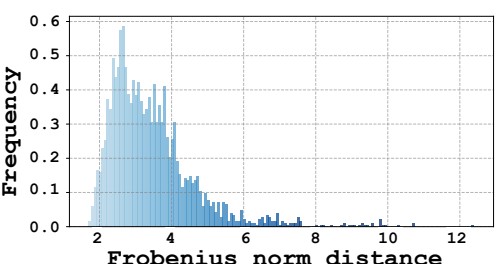

Figure 1: **Temporal Redundancy:** Distribution of the distance between the feature of tokens in the previous and the current timestep.

Figure 2: **Error Propagation:** Distribution of the error in the final layer output when the same noise is applied to each token in the first layer.

**Difference in Error Propagation:** Figure 2 introduces the other interesting perspective of *error propagation* in diffusion transformers. Specifically, self-attention and cross-attention layers are widely utilized in diffusion transformers to formulate the dependency between different tokens. As a result, the error in one of the tokens may propagate to some other tokens by self-attention and finally result in the error in all of the tokens. To understand the error propagation in tokens of diffusion transformers, we apply Gaussian noise with the same intensity to each token and compute the resulting error in all the tokens in the output of the final layer. Surprisingly, Figure 2 shows that the same noise in different tokens leads to significantly different propagation errors, with the largest propagation error being more than $10\times$ the smallest one. In the context of feature caching, this indicates that *the same error introduced by feature cache can result in vastly different errors in the final generation result due to error propagation.*

Moreover, we have also investigated the difference of the tokens in layers of different depths and types, which demonstrates significant differences, as introduced in the following sections. In summary, different tokens exhibit *significant differences* in their sensitivities to feature caching, indicating that they deserve different priorities during the caching process. This motivates us to study the token-wise feature caching strategy, which aims to select the maximal number of tokens to maximize the acceleration ratio while minimizing the resulting error introduced by the feature caching by selecting the tokens that make the least caching error.

To tackle this challenge, this paper introduces Token-wise feature Caching `ToCa` for training-free acceleration of diffusion transformers, which provides a fine-grained caching strategy for different tokens in the same layer and the tokens in different layers. The core challenge of `ToCa` is to accurately select the tokens that are suitable for feature caching with the computation-cheapest operations. Consistent with the two previous analyses, we mainly study this problem from the perspective of *temporal redundancy* and *error propagation* by defining four scores for token selection. Specifically, for *temporal redundancy*, we try to select the tokens with the highest similarity (*i.e.* lowest difference) with their value in the previous timesteps by considering their frequency of being cached, as well as their distribution in the spatial dimension of the images. For *error propagation*, we attempt to cache the tokens which makes the least influence on other tokens based on their attendance in self-attention and cross-attention layers. Besides, all of these scores can be obtained without any additional computation costs.

Extensive experiments on text-to-image, text-to-video and class-to-image generation demonstrate the effectiveness of `ToCa` on PixArt-$\alpha$, OpenSora, and DiT over previous feature caching methods. For instance, **2.36×** acceleration can be achieved on OpenSora without requirements for training, outperforming halving the number of timesteps directly by 1.56 on VBench. On PariPrompt, `ToCa` even leads to 1.13 improvements on CLIP Score, indicating higher consistency to the text conditions.

In summary, the contributions of this paper are as follows:

1. We propose Token-wise Caching (`ToCa`) as a fine-grained feature caching strategy tailored to acceleration for diffusion transformers. To the best of our knowledge, `ToCa` first introduces the perspective of error propagation in feature caching methods.
2. We introduce four scores to select the most suitable tokens for feature caching in each layer with no additional computation costs. Besides, `ToCa` enables us to apply different caching ratios in layers of different depths and types, and also bring a bag of techniques for feature caching.
3. Abundant experiments on PixArt-$\alpha$, OpenSora, and DiT have been conducted, which demonstrates that `ToCa` achieves a high acceleration ratio while maintaining nearly lossless generation quality. Our codes have been released for further exploration in this domain.

## 2 RELATED WORK

**Transformers in Diffusion Models**

Diffusion models (DMs) (Ho et al., 2020; Sohl-Dickstein et al., 2015), which iteratively denoise an initial noise input through a series of diffusion steps, have achieved remarkable success across various generation applications (Rombach et al., 2022; Balaji et al., 2022). Early DMs (Ho et al., 2020; Rombach et al., 2022) are based on the U-Net architecture (Ronneberger et al., 2015), consistently achieving satisfactory generation results. Recently, Diffusion Transformer (DiT) (Peebles & Xie, 2023) has emerged as a major advancement by replacing the U-Net backbone with a Transformer architecture. This transition enhances the scalability and efficiency of DMs across various generative tasks (Chen et al., 2024a; Brooks et al., 2024). For example, PixArt-$\alpha$ (Chen et al., 2024a) utilizes DiT as a scalable foundational model, adapting it for text-to-image generation, while Sora (Brooks et al., 2024) demonstrates DiT's potential in high-fidelity video generation, inspiring a series of related open-source projects (Zheng et al., 2024; Lab & etc., 2024). Despite their success, the iterative denoising process of these DMs is significantly time-consuming, making them less feasible for practical applications.

**Acceleration of Diffusion Models**

To improve the generation efficiency of DMs, numerous diffusion acceleration methods have been proposed, falling broadly into two categories: (1) *reducing the number of sampling timesteps*, and (2) *accelerating the denoising networks*. The first category aims to achieve high-quality generation results with fewer sampling steps. DDIM (Song et al., 2021) introduces a deterministic sampling process that reduces the number of denoising steps while preserving generation quality. DPM-Solver (Lu et al., 2022a) and DPM-Solver++ (Lu et al., 2022b) propose adaptive high-order solvers for a faster generation without compromising on generation results. Rectified flow (Liu et al., 2023) optimizes distribution transport in ODE models to facilitate efficient and high-quality generation, enabling sampling with fewer timesteps. Step-distillation (Salimans & Ho, 2022; Meng et al., 2023) minimizes the number of timesteps with knowledge distillation from multiple timesteps to fewer ones. Consistency models (Song et al., 2023) accelerate generative modeling by mapping noise directly to data and enforcing self-consistency across steps. In the second category, various efforts have been paid to token reduction (Bolya & Hoffman, 2023; Zhang et al., 2025; 2024a), knowledge distillation (Li et al., 2024), and weight quantization (Li et al., 2023b; Shang et al., 2023) and pruning (Fang et al., 2023) on the denoising networks. Additionally, recent cache-based methods reduce redundant computations to accelerate inference for DMs. These cache-based methods have obtained abundant attention since they have no requirements for additional training. DeepCache (Ma et al., 2024b) eliminates redundant computations in Stable Diffusion (Rombach et al., 2022) by reusing intermediate features of low-resolution layers in the U-Net. Faster Diffusion (Li et al., 2023a) accelerates the sampling process of DMs by caching U-Net encoder features across timesteps, skipping encoder computations at certain steps. Unfortunately, DeepCache and Faster Diffusion are designed specifically for U-Net-based denoisers and can not be applied to DiT (Chen et al., 2024b). Recently, FORA (Selvaraju et al., 2024) and $\Delta$-DiT (Chen et al., 2024b) have been proposed to cache the features and the residual of features for DiT. Learning-to-Cache (Ma et al., 2024a) learns an optimal cache strategy, which achieves a slightly higher acceleration ratio but introduces the requirements of training. However, these methods apply the identical cache solution to all the tokens and even all the layers, which leads to a significant performance degradation in generation quality.

## 3 METHODOLOGY

In the Methodology section, we briefly introduce Diffusion Models and the feature caching acceleration method for Diffusion Transformers, followed by the `ToCa` workflow and important token selection. The importance of a token $x_i$ is determined by $s_1$ **interaction strength with other tokens**, $s_2$ **association with global textual information**, $s_3$ **accumulated cache error**, which, if excessive, can cause image collapse and should be controlled, and $s_4$ **spatial uniformity** of selected tokens.

### 3.1 PRELIMINARY

**Diffusion Models** Diffusion models are formulated to contain two processes, including a forward process which adds Gaussian noise to a clean image, and a reverse process which gradually denoises a standard Gaussian noise to a real image. By denoting $t$ as the timestep and $\beta_t$ as the noise variance

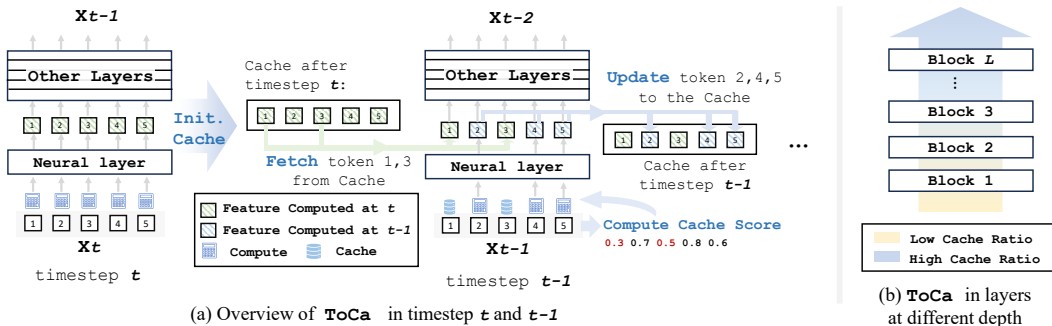

(a) Overview of `ToCa` in timestep *t* and *t-1*

(b) `ToCa` in layers at different depth

Figure 3: The overview of ToCa on the example of the first layer with caching ratio $R = 40\%$. (a) In the first timestep of the cache period, ToCa computes all the tokens and stores them in the cache for initialization. Then, in the next timestep, ToCa first computes the caching score of each token and selects the tokens for cache based on them. Then, ToCa fetches the features of cached tokens from the cache while performing real computations in the other tokens. Then, the features of tokens that have been computed are utilized to update their value in the cache. (b) ToCa applies a higher cache ratio in the deep layer and a relatively lower cache ratio in the shallow layers.

schedule, then the conditional probability in the reverse (denoise) process can be modeled as

$$p_\theta(x_{t-1} \mid x_t) = \mathcal{N}\left(x_{t-1}; \frac{1}{\sqrt{\alpha_t}}\left(x_t - \frac{1-\alpha_t}{\sqrt{1-\bar{\alpha}_t}}\epsilon_\theta(x_t, t)\right), \beta_t \mathbf{I}\right), \tag{1}$$

where $\alpha_t = 1 - \beta_t$, $\bar{\alpha}_t = \prod_{i=1}^{T} \alpha_i$, and $T$ denotes the number of timesteps. Importantly, $\epsilon_\theta$ denotes a denoising network with its parameters $\theta$ that takes $x_t$ and $t$ as the input and then predicts the corresponding noise for denoising. For image generation with $\mathcal{T}$ timesteps, $\epsilon_\theta$ is required to infer for $\mathcal{T}$ times, which takes most of the computation costs in the diffusion models. Recently, fruitful works demonstrate that formulating $\epsilon_\theta$ as a transformer usually leads to better generation quality.

**Diffusion Transformer** Diffusion transformer models are usually composed of stacking groups of self-attention layers $f_{\text{SA}}$, multilayer perceptron $f_{\text{MLP}}$, and cross-attention layers $f_{\text{CA}}$ (for conditional generation). It can be roughly formulated as $g_1 \circ g_2 \circ \ldots g_L$ where $g^i = \{f_{\text{SA}}^i, f_{\text{CA}}^i, f_{\text{MLP}}^i\}$. The upper script denotes the index of layer groups and $L$ denotes its maximal number. We omit the other components such as layer norm and residual connections here for simplicity. For diffusion transformers, the input data $\mathbf{x}_t$ is a sequence of tokens corresponding to different patches in the generated images, which can be formulated as $\mathbf{x}_t = \{x_i\}_{i=1}^{H \times W}$, where $H$ and $W$ denote the height and width of the images or the latent code of images, respectively.

## 3.2 NAIVE FEATURE CACHING FOR DIFFUSION TRANSFORMERS

We follow the naive scheme for feature caching adopted by most previous caching methods(Ma et al., 2024b) for diffusion denoisers. Given a set of $\mathcal{N}$ adjacent timesteps $\{t, t+1, \ldots, t+\mathcal{N}-1\}$, native feature caching performs the complete computation at the first timestep $t$ and stores the intermediate features in all the layers, which can be formulated as $\mathcal{C}(\mathbf{x}_t) := f(\mathbf{x}_t^l)$, for $\forall l \in [0, L]$, where ":=" indicates the operation of assigning the value. Then, in the next $\mathcal{N}-1$ timesteps, feature caching avoids the computation of self-attention, cross-attention, and MLP layers by reusing the feature cached at timestep $t$. By denoting the cache as $\mathcal{C}$ and the expected feature of the input $\mathbf{x}_t$ in the $l_{th}$ layer as $\mathcal{F}(\mathbf{x}_t^l)$, then for $\forall l \in [1, L]$, the naive feature caching can be formulated as

$$\mathcal{F}(\mathbf{x}_{t+1}^{l-1}) = \mathcal{F}(\mathbf{x}_{t+2}^{l-1}) = \cdots = \mathcal{F}(\mathbf{x}_{t+\mathcal{N}}^{l-1}) := \mathcal{C}(\mathbf{x}_t^l). \tag{2}$$

In these $\mathcal{N}$ timesteps, naive feature caching avoids almost all the computation in $\mathcal{N}-1$ timesteps, leading to around $\mathcal{N}-1$ times acceleration. After the $\mathcal{N}$ timesteps, the feature cache then starts a new period from initializing the cache as aforementioned, again. The effectiveness of feature caching can be explained by the extremely low difference between the tokens in the adjacent timesteps. However, as $\mathcal{N}$ increases, the difference between the feature value in the cache and their correct value can be exponentially increased, leading to degeneration in generation quality, which motivates us to study more fine-grained methods for feature caching.

## 3.3 TOKEN-WISE FEATURE CACHING

The naive feature caching scheme caches all the tokens of the diffusion transformers with the same strategy. However, as demonstrated in Figure 1, 2 and 5, feature cache introduces significantly

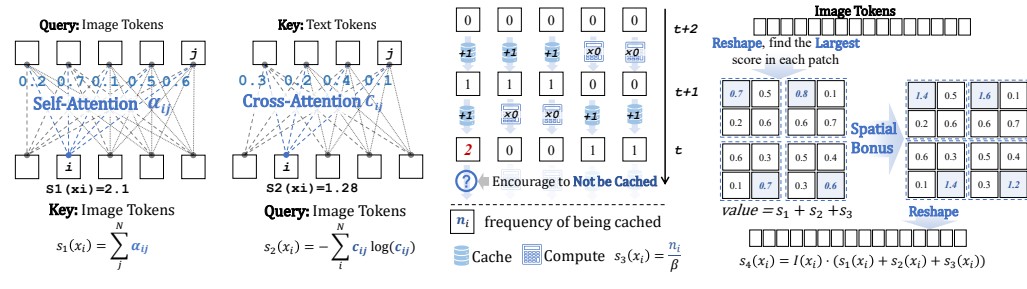

**(I) Influence to Other Tokens**    **(II) Control Ability**    **(III) Cache Frequency**    **(IV) Uniform Spatial Distribution**

Figure 4: The computation of caching scores in `ToCa`, where a token with a **lower cache score** is encouraged to be **cached**: **(I)** Self-attention weights are utilized to measure the influence of each token on the other tokens, where a token with higher influence is considered not suitable for caching. **(II)** Cross-attention weights are utilized to measure the influence of each image token on the text (condition) tokens, where an image token with higher entropy is considered not suitable for caching. **(III)** Tokens that have been cached multiple times are encouraged to not be cached in the following layers. **(IV)** We increase the cache score for the token with the largest cache score in its neighboring pixels to make the cached tokens distributed uniformly in the spatial dimension.

different influence to different tokens, motivating us to design a more fine-grained caching method in the token-level. In this section, we begin with the overall framework of `ToCa`, then introduce our strategy for token selection and caching ratios.

### 3.4 OVERALL FRAMEWORK

**Cache Initialization** Similar to previous caching methods, given a set of adjacent timesteps $\{t, t+1, \ldots, t+\mathcal{N}-1\}$, our method begins with computing all the tokens at the first timestep $t$, and storing the computation result (intermediate features) of each self-attention, cross-attention, and MLP layer in a cache, denoted by $\mathcal{C}$, as shown in the left part in Figure 3. This can be considered as the initialization of $\mathcal{C}$, which has no difference compared with previous caching methods.

**Computing with the Cache** In the following timesteps, we can skip the computation of some unimportant tokens by re-using their value in the cache $\mathcal{C}$. We firstly pre-define the *cache ratio $R$* of tokens in each layer, which indicates that the computation of $R\%$ of the tokens in this layer should be skipped by using their value in the cache, and the other $(1-R\%)$ tokens should still be computed. To achieve this, a caching score function $\mathcal{S}$ is introduced to decide whether a token should be cached, which will be detailed in the next section. Then, with $\mathcal{S}$, we can select a set of cached tokens as $\mathcal{I}_{\text{Cache}}$ and the other set of tokens for real computation as $\mathcal{I}_{\text{Compute}}=\{x_i\}_{i=1}^N - \mathcal{I}_{\text{Cache}}$. Then, the computation of the layer $f$ for $i_{th}$ token $x_i$ can be formulated as $\mathcal{F}(x_i) = \gamma_i f(x_i) + (1-\gamma_i)\mathcal{C}(x_i)$, where $\gamma_i = 0$ for $i \in \mathcal{I}_{\text{Cache}}$ and $\gamma_i = 1$ for $i \in \mathcal{I}_{\text{Compute}}$. $\mathcal{C}(x_i)$ denotes fetching the cached value of $x_i$ from $\mathcal{C}$, which has no computation costs and hence leads to overall acceleration in $f$.

**Cache Updating** As a significant difference between traditional cache methods and `ToCa`, traditional cache methods only update the feature in the cache at the first timestep for each caching period while `ToCa` can update the feature in the cache at all the timesteps, which helps to reduce the error introduced by feature reusing. For the tokens $x_i \in \mathcal{I}_{\text{Compute}}$ which are computed, we update their corresponding value in the cache $\mathcal{C}$, which can be formulated as $\mathcal{C}(x_i) := \mathcal{F}(x_i)$ for $i \in \mathcal{I}_{\text{Compute}}$.

### 3.5 TOKEN SELECTION

Given a sequence of tokens $\mathbf{x}_t = \{x_i\}_{i=1}^N$, token selection aims to select the tokens that are suitable for caching. To this end, we define a caching score function $\mathcal{S}(x_i)$ to decide whether the $i_{th}$ token $x_i$ should be cached, where a token with a higher score has a lower priority for caching and a higher priority to be actually computed. The $\mathcal{S}(x_i)$ is composed of four sub-scores $\{s_1, s_2, s_3, s_4\}$, corresponding to the following four principals.

**(I) Influence to Other Tokens:** If a token has a significant contribution to the value of other tokens, then the error caused by token caching on this token can easily propagate to the other tokens, ultimately leading to discrepancies between all tokens and their correct values. Consequently, we consider the contribution of each token to other tokens as one of the criteria for defining whether it should be cached, estimated with an attention score in self-attention. Recall that the self-attention can be formulated as $\mathbf{O} = \mathbf{A}\mathbf{V}$, where $\mathbf{A} = \text{Softmax}(\frac{\mathbf{Q}\mathbf{K}^{\mathbf{T}}}{\sqrt{\mathbf{d}}}) \in \mathbb{R}^{N \times N}$ denotes the normalized

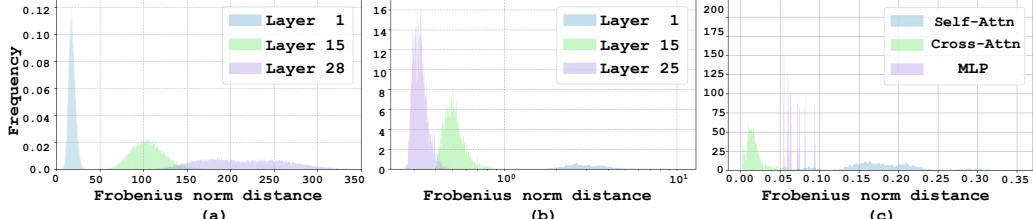

Figure 5: (a) The distance between features at the last timestep and the current timestep for features in different layer depths. (b) The distribution of errors in the output of the final layer when the same Gaussian noise is applied to tokens in different layer depths. (c) The distribution of errors in the output of the final layer when the same Gaussian noise is applied to tokens in different layer types.

attention map. $\mathbf{Q}$, $\mathbf{K}$, $\mathbf{V}$ and $\mathbf{O} \in \mathbb{R}^{N \times d}$ are query, key, value and output tokens respectively; $\mathbf{d}$ is the hidden size of each token and $N$ is the total number of tokens. More specifically, the $i_{th}$ output token is obtained through $o_i = \sum_{j=1}^{N} \alpha_{ij} v_j$, where $\alpha_{ij}$ is the $(i, j)$ element of the attention map $\mathbf{A}$, denoting the contribution of value token $v_j$ to the output token $o_j$. With these notations, we define $s_1$ to measure the contribution of $x_i$ to other tokens as $s_1(x_i) = \lambda_1 \sum_{i=1}^{N} \alpha_{ij}$, as shown in Figure 4(I).

**(II) Influence to Control Ability:** The control ability of diffusion models in the text-to-image generation is usually achieved with a cross-attention layer which injects the control signal (*e.g.* text) into the image tokens. Hence, the cross-attention map reflects how each image token is influenced by the control signal. In this paper, we define the image tokens that are influenced by more tokens in the control signal as the tokens that are not suitable for caching, since the caching error on these tokens leads to more harm in the control ability. Specifically, by denoting $c_{ij}$ as the $(i, j)$ element in the cross attention score $\mathbf{C} = \text{Softmax}(\frac{\mathbf{Q}\mathbf{K}_{\text{text}}^{\mathbf{T}}}{\sqrt{\mathbf{d}}})$, where $\mathbf{K}_{\text{text}}$ denotes the keys of text tokens (control tokens). Then, as shown in Figure 4(II), we employ the entropy $H(x_i)$ of the cross-attention weight for each image token $x_i$ as its influence on the control-ability of diffusion models, which can be formulated as $s_2(x_i) = H(x_i) = -\sum_{j=1}^{N} c_{ij} \log(c_{ij})$.

**(III) Cache Frequency:** We observe that when a token is cached across multiple adjacent layers, the error introduced by feature caching in this token can be quickly accumulated, and the difference between it and its correct value can be exponentially amplified, which significantly degrades the overall quality of images. Hence, we define recently cached tokens as unsuitable for cache in the next layers and time steps. Conversely, the tokens that have not been cached for multiple layers and timesteps are encouraged to be cached. As shown in Figure 4(III), this selection rule is achieved by recording the times of being cached for each token after their last real computation, which can be formulated as $s_3(x_i) = \frac{n_i}{\mathcal{N}}$, where $n_i$ represents the number of times that $x_i$ has been cached after its last computation. $\mathcal{N}$ is the number of timesteps in each feature caching cycle.

**(IV) Uniform Spatial Distribution:** The pixels in the neighboring patch of the images usually contain similar information. As discussed in previous works, overwhelmingly pruning the information in a local spatial region may result in significant performance degradation in the whole images (Bolya & Hoffman, 2023). Hence, to guarantee that the errors introduced by caching are not densely distributed in the same spatial region, we define the following scoring function: $s_4(x_i) = \mathcal{I}(x_i) \cdot (\lambda_1 \cdot s_1(x_i) + \lambda_2 \cdot s_2(x_i) + \lambda_3 \cdot s_3(x_i))$, where $\mathcal{I}(x_i)$ is an indicator function which equals to 1 if $x_i$ has the highest score of $\lambda_1 \cdot s_1(x_i) + \lambda_2 \cdot s_2(x_i) + \lambda_3 \cdot s_3(x_i)$ in its neighboring $k \times k$ pixels and 0 in the other settings, and $\lambda_j$ are hyper-parameters to balance each score.

**In summary**, the overall caching score of $x_i$ can be formulated as $\mathcal{S}(x_i) = \sum_{j=1}^{4} \lambda_j \cdot s_j(x_i)$, where $\lambda_j$ are hyper-parameters to balance each score. Then, with the cache ratio $R$, the index set for the cached tokens is obtained in the following form:

$$\mathcal{I}_{\text{Cache}} = \underset{\{i_1, i_2, \ldots, i_{R\% \times N}\} \subseteq \{1, 2, \ldots, n\}}{\arg\min} \{\mathcal{S}(x_{i_1}), \mathcal{S}(x_{i_2}), \cdots, \mathcal{S}(x_{i_n})\}. \tag{3}$$

### 3.6 DIFFERENT CACHE RATIOS IN DIFFERENT LAYERS

Figure 5 shows the difference in feature caching of different layers, where (a) shows that the output features of different layers have different distances compared with their value in the last step (*i.e.* different temporal redundancy). (b) and (c) show that when errors in the same density are applied to

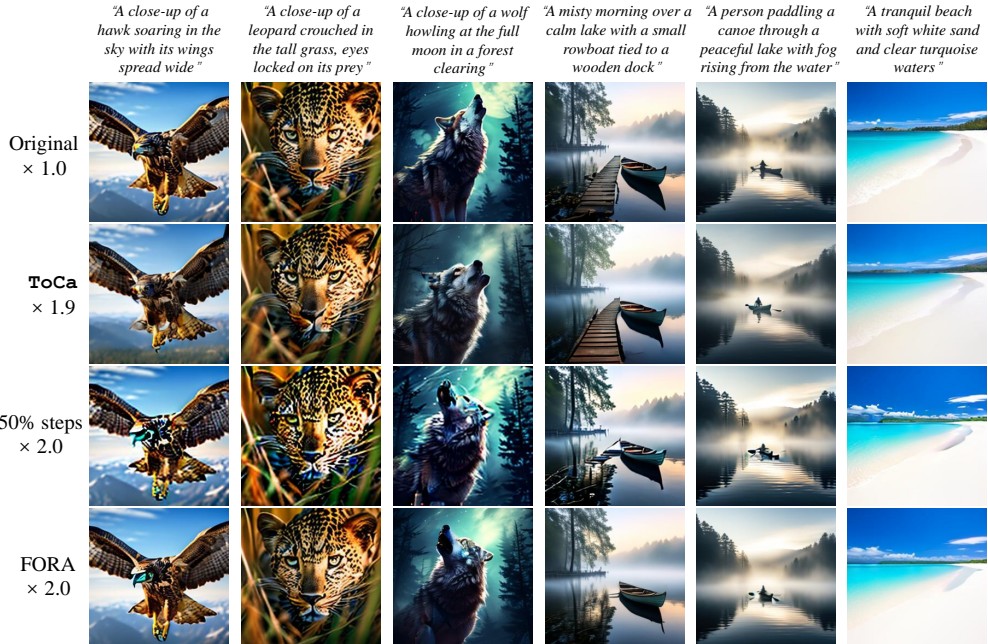

Figure 6: Visualization examples for different acceleration methods on PixArt-$\alpha$.

layers in different depths and types, the resulting error in the final layer exhibits extremely different magnitudes. Specifically, in the three studies, the disparity between the maximum and minimum error can be several orders of magnitude. Fortunately, ToCa enables us to apply different caching ratios for layers in various depths and types. By denoting the overall caching ratio for all the layers and timesteps as $\mathcal{R}_0$, then two factors $r_{\text{depth}}$ and $r_{\text{type}}$ are introduced to adjust the caching ratios. Then the final caching ratio of the layer in $l$ depth and type can be written as $R^l_{\text{type}} = R \times r_l \times r_{\text{type}}$.

$r_l$: As introduced in Figure 5(a) and (b) , although the features of the shallow layers tend to exhibit lower differences than the deeper layers, the error introduced by the cached tokens in the shallow layers can be propagated to the other tokens and amplified during the computation in all the following layers, resulting in a much larger caching error. Based on this observation, we set larger and smaller cache ratios for deeper and shallower layers, respectively, by setting $r_l = 0.5 + \lambda_l(l/L - 0.5)$, where 0.5 is utilized for 1-center and $L$ denotes the maximal depth and $\lambda_l$ controls the slope.

$r_{\text{type}}$: As shown in Figure 5(c), layers of different types have different sensitivities to feature caching. Especially, the error on the token in self-attention layers can quickly propagate to other tokens, due to the property that each token in self-attention layers can attend to all the tokens. A naive solution is to set lower cache ratios for self-attention layers. However, we observe that even if only a smaller ratio of tokens is cached, the error introduced by these tokens still quickly propagates to all other tokens, and has almost the same negative influence as caching all the tokens. Based on this fact, we propose to cache all tokens in self-attention layers. For MLP and cross-attention layers, $r_{\text{type}}$ is set to the ratio of their computation costs over the overall computation costs. This strategy encourages layers with more computation costs to have a higher cache ratio.

## 4 EXPERIMENT

### 4.1 EXPERIMENT SETTINGS

**Model Configurations** We conduct experiments on three commonly-used DiT-based models across different generation tasks, including PixArt-$\alpha$ (Chen et al., 2024a) for text-to-image generation, OpenSora (Zheng et al., 2024) for text-to-video generation, and DiT-XL/2 (Peebles & Xie, 2023) for class-conditional image generation with NVIDIA A800 80GB GPUs. Each model utilizes its default sampling method: DPM-Solver++ (Lu et al., 2022b) with 20 steps for PixArt-$\alpha$, rflow (Liu et al., 2023) with 30 steps for OpenSora and DDPM (Ho et al., 2020) with 250 steps for DiT-XL/2. For each model, we configure different average forced activation cycles $\mathcal{N}$ and average caching ratios $R$ for ToCa as follows: PixArt-$\alpha$: $\mathcal{N} = 3$ and $R = 70\%$, OpenSora: $\mathcal{N} = 3$ for temporal

Table 1: **Qualitative comparison of text-to-image generation** on MS-COCO2017 and PartiPrompts with PixArt-$\alpha$ and 20 DPM++ sampling steps by default.

| Method | Latency(s) ↓ | FLOPs ↓ | Speed ↑ | MS-COCO2017 | | PartiPrompts |
| | | | | FID-30k ↓ | CLIP ↑ | CLIP ↑ |
|---|---|---|---|---|---|---|
| **PixArt-$\alpha$** (Chen et al., 2024a) | 0.682 | 11.18 | 1.00× | 28.09 | 16.32 | 16.70 |
| 50% **steps** | 0.391 | 5.59 | 2.00× | 37.46 | 15.85 | 16.37 |
| **FORA**($\mathcal{N} = 2$) (Selvaraju et al., 2024) | 0.416 | 5.66 | 1.98× | 29.67 | 16.40 | 17.19 |
| **FORA**($\mathcal{N} = 3$) (Selvaraju et al., 2024) | 0.342 | 4.01 | 2.79× | 29.88 | 16.42 | 17.15 |
| **ToCa** ($\mathcal{N} = 3, R = 60\%$) | 0.410 | 6.33 | 1.77× | **28.02** | **16.45** | 17.15 |
| **ToCa** ($\mathcal{N} = 3, R = 70\%$) | 0.390 | 5.78 | 1.93× | 28.33 | 16.44 | 17.75 |
| **ToCa** ($\mathcal{N} = 3, R = 80\%$) | 0.370 | 5.05 | 2.21× | 28.82 | 16.44 | **17.83** |
| **ToCa** ($\mathcal{N} = 3, R = 90\%$) | 0.347 | 4.26 | 2.62× | 29.73 | 16.45 | 17.82 |

Table 2: **Quantitative comparison in text-to-video generation** on VBench. *Results are from PAB (Zhao et al., 2024). PAB$^{1-3}$ indicate PAB with different hyper-parameters.

| Method | Latency(s) ↓ | FLOPs(T) ↓ | Speed ↑ | VBench(%) ↑ |
|---|---|---|---|---|
| **OpenSora** (Zheng et al., 2024) | 81.18 | 3283.20 | 1.00× | 79.13 |
| $\Delta$-**DiT**$^*$ (Chen et al., 2024b) | 79.14 | 3166.47 | 1.04× | 78.21 |
| **T-GATE**$^*$ (Zhang et al., 2024b) | 67.98 | 2818.40 | 1.16× | 77.61 |
| **PAB**$^{1*}$ (Zhao et al., 2024) | 60.78 | 2657.70 | 1.24× | 78.51 |
| **PAB**$^{2*}$ (Zhao et al., 2024) | 59.16 | 2615.15 | 1.26× | 77.64 |
| **PAB**$^{3*}$ (Zhao et al., 2024) | 56.64 | 2558.25 | 1.28× | 76.95 |
| 50% **steps** | 42.72 | 1641.60 | 2.00× | 76.78 |
| **FORA**(Selvaraju et al., 2024) | 49.26 | 1751.32 | 1.87× | 76.91 |
| **ToCa**($R = 80\%$) | 43.52 | 1439.70 | 2.28× | **78.59** |
| **ToCa**($R = 85\%$) | 43.08 | 1394.03 | **2.36×** | 78.34 |

attention, spatial attention, MLP, and $\mathcal{N} = 6$ for cross-attention, with $R = 85\%$ exclusively for MLP, and DiT: $\mathcal{N} = 4$ and $R = 93\%$. Please refer to the appendix for more implementation details.

**Evaluation and Metrics** For text-to-image generation, we utilize 30,000 captions randomly selected from COCO-2017 (Lin et al., 2014) to generate an equivalent number of images. FID-30k is computed to assess image quality, while the CLIP Score (Hessel et al., 2021) is used to evaluate the alignment between image content and captions. In the case of text-to-video generation, we leverage the VBench framework (Huang et al., 2024), generating 5 videos for each of the 950 benchmark prompts under different random seeds, resulting in a total of 4,750 videos. The generated videos are comprehensively evaluated across 16 aspects proposed in VBench. For class-conditional image generation, we uniformly sample from 1,000 classes in ImageNet (Deng et al., 2009) to produce 50,000 images at a resolution of $256 \times 256$, evaluating performance using FID-50k (Heusel et al., 2017). Additionally, we employ sFID, Precision, and Recall as supplementary metrics.

## 4.2 RESULTS ON TEXT-TO-IMAGE GENERATION

In Table 1, we compare `ToCa` configured with parameters to achieve an acceleration ratio close to 2.0, against two other training-free acceleration approaches: FORA (Selvaraju et al., 2024), a recent cache-based high-acceleration method, and the 10-step DPM-Solver++ sampling (Lu et al., 2022b). In terms of ***generation quality***, the quantitative results demonstrate that `ToCa` achieves the lowest FID among the compared acceleration methods while maintaining a high acceleration ratio. Figure 6 also illustrates that our generated results most closely resemble those of the original PixArt-$\alpha$. Regarding ***generation consistency***, Table 1 demonstrates that `ToCa` achieves the highest CLIP score on both MS-COCO2017(Lin et al., 2014) and the PartiPrompts(Yu et al., 2022). Figure 6 shows that `ToCa` generates images that align more closely with the text descriptions compared to other methods. This is particularly evident in the fourth case, where only `ToCa` successfully generates an image matching *"a small rowboat tied to a wooden dock"*, while other methods fail to generate the content of *"a wooden dock"*. This may be caused by cross-attention score $s_2$ in `ToCa` that ensures the frequent refreshing of tokens that are semantically relevant to the text descriptions, resulting in generated images with enhanced semantic consistency to the text prompts.

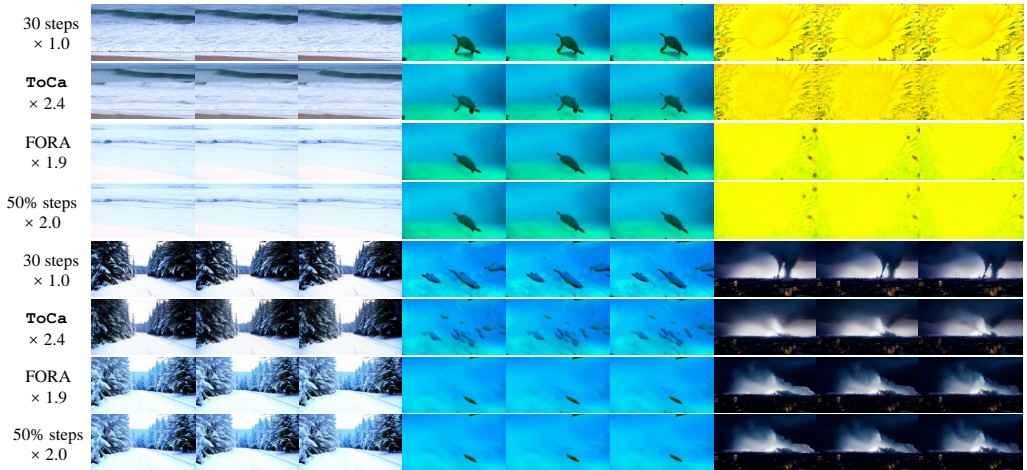

Figure 7: Visualization examples for different acceleration methods on OpenSora. Please kindly refer to the supplementary material or our web page for viewing these videos.

Notably, Table 1 shows that under similar acceleration ratios, ToCa exhibits a very marginal decrease in generation quality. In contrast, directly halving the number of timesteps leads to 9.37 increments in FID, indicating a significant performance drop. This observation indicates that when the number of sampling steps is already relatively low (*e.g.*, 20 steps), further reduction in the number of sampling steps may severely compromise the generation quality. In contrast, ToCa remains effective, demonstrating the distinct advantage of ToCa in the situation of low sampling steps.

### 4.3 RESULTS ON TEXT-TO-VIDEO GENERATION

We compare ToCa with adjusted rflow sampling steps from 20 to 10, alongside other acceleration methods including FORA, PAB (Zhao et al., 2024), Δ-DiT (Chen et al., 2024b), and T-GATE (Zhang et al., 2024b) using OpenSora (Zheng et al., 2024) for text-to-video generation. As presented in Table 2, the experimental results show that ToCa achieves an impressive VBench score offering the lowest computational cost and highest inference speed among all methods tested. The VBench score of the $2.36\times$ accelerated ToCa scheme drops by only 0.79 compared to the non-accelerated scheme, while FORA's score decreases by 2.22, resulting in a 64.4% reduction in quality loss. Additionally, more VBench metrics results are presented in Figure 8, which illustrate that ToCa significantly speeds up

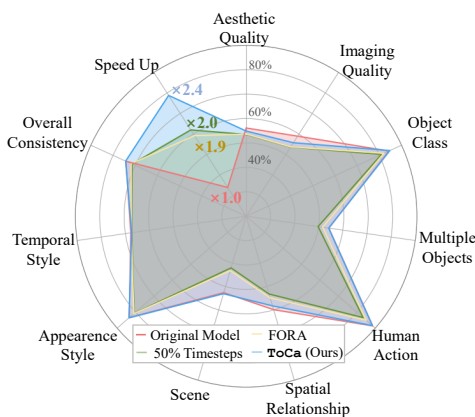

Figure 8: VBench metrics and acceleration ratio of proposed **ToCa** and other methods

the original OpenSora with only slight performance degradation on a few metrics. Notably, ToCa stands out as the sole acceleration method achieving nearly overall consistency performance with the original OpenSora, clearly outperforming another cache-based acceleration method FORA. This again highlights the effectiveness of our proposed cross-attention-based token selection strategy, ensuring that the generated videos are highly aligned with the text descriptions. We further present some video generation results in Figure 7, where we observe that the visual fidelity, and overall consistency of ToCa are closest to the original OpenSora. Please kindly refer to our video demos in the supplementary material or the web page for viewing the video demo.

### 4.4 RESULTS ON CLASS-CONDITIONAL IMAGE GENERATION

Quantitative comparison between ToCa with other training-free DiT acceleration methods is shown in Table 3, which demonstrates that ToCa outperforms other methods in terms of both FID and sFID by a clear margin under the similar acceleration ratio. For instance, ToCa leads to 0.39 and 0.22 lower values in sFID and FID compared with FORA with a similar acceleration ratio, respectively.

Table 3: **Quantitative comparison on class-to-image generation** on ImageNet with DiT-XL/2.

| Method | Latency(s) ↓ | FLOPs(T) ↓ | Speed ↑ | sFID ↓ | FID ↓ | Precision ↑ | Recall ↑ |
|---|---|---|---|---|---|---|---|
| **DiT-XL/2-G (cfg** $= 1.50$**)** | 2.012 | 118.68 | $1.00\times$ | 4.98 | 2.31 | 0.82 | 0.58 |
| $33\%$ **steps** | 0.681 | 39.40 | $3.01\times$ | 6.31 | 2.76 | 0.81 | 0.57 |
| $37\%$ **steps** | 0.748 | 44.15 | $2.69\times$ | 6.04 | 2.64 | 0.81 | 0.58 |
| **FORA**($\mathcal{N} = 3$) | 0.807 | 39.95 | $2.97\times$ | 6.21 | 2.80 | 0.80 | 0.59 |
| **FORA**($\mathcal{N} = 2.8$) | 0.815 | 43.36 | $2.74\times$ | 6.13 | 2.80 | 0.80 | 0.59 |
| **ToCa** ($\mathcal{N} = 4, R = 93\%$) | 0.820 | 43.22 | $2.75\times$ | **5.74** | **2.58** | **0.81** | **0.59** |

Table 4: **Ablation studies** with DiT-XL/2-G (cfg-1.50). $s_1$ is used in all experiments. $s_2$ is not used since DiT does not have cross-attention layers.

| $R$ Schedule | | Uniform Spatial | Cache | ImageNet |
|---|---|---|---|---|
| $r_l$ | $r_{type}$ | Distribution $s_4$ | Frequency $s_3$ | FID-5k ↓ |
| ✓ | ✓ | ✓ | ✓ | **9.32** |
| ✗ | ✓ | ✓ | ✓ | 9.60 |
| ✓ | ✗ | ✓ | ✓ | 9.67 |
| ✓ | ✓ | ✗ | ✓ | 9.35 |
| ✓ | ✓ | ✓ | ✗ | 9.65 |

Table 5: **Ablation studies** of token selection based on different attention scores ($s_1$ and $s_2$) with PixArt$-\alpha$. "Random" indicates replacing attention scores with random values. $s_3, s_4, r_l, r_{type}$ are used in all three settings.

| Token Selection Methods | MS-COCO2017 FID-30k ↓ | PartiPrompts CLIP ↑ |
|---|---|---|
| Cross-Attention $s_2$ | 28.33 | **17.75** |
| Self-Attention $s_1$ | **28.21** | 17.13 |
| Random | 28.46 | 17.08 |

**Ablation Study** Table 4 presents the effect of the two factors on adjusting the caching ratio in different layers, where applying different caching ratios to layers in different types ($r_{type}$) and different depths ($r_l$) leads to 0.28 and 0.35 FID reduction, respectively. Besides, using the score of uniform spatial distribution ($s_3$) and cache frequency ($s_4$) reduces FID by 0.02 and 0.33, respectively. Table 5 compares the influence of selecting tokens with the self-attention weights ($s_1$) and the cross-attention weights ($s_2$). The other ToCa modules including $s_3, s_4, r_l, r_{type}$ are utilized in these experiments. It is observed that $s_1$ tends to achieve a lower FID while $s_2$ tends to reach a higher CLIP score, which is reasonable since self-attention is mainly utilized for the generation of the overall images while cross-attention is utilized to inject the conditional signals. In summary, these results demonstrate that all the cache scores in ToCa have their benefits in different dimensions.

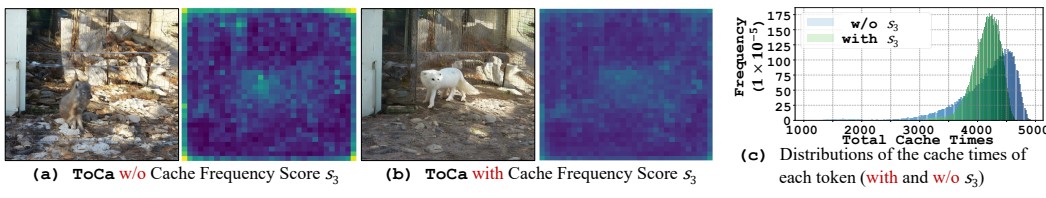

(a) **ToCa** w/o Cache Frequency Score $s_3$     (b) **ToCa** with Cache Frequency Score $s_3$     (c) Distributions of the cache times of each token (with and w/o $s_3$)

Figure 9: Visualization of cached tokens selected with and without $s_3$ (cache frequency). The pixel with a darker color indicates the corresponding tokens are more frequently cached. (c) The distribution of the number of being cached for each token w/ and w/o $s_3$.

**Visualization on the Cached Tokens** Figure 9 (a-b) show the number of times that each token is cached during generation, where darker colors indicate more frequent caching. It is observed that both the two schemes perform more cache in the unimportant background tokens while performing more real computations in the tokens of the Arctic fox. However, the image without $s_3$ has a bad quality in the background since the background tokens have been cached too many times. In contrast, applying the score of cache frequency $s_3$, which aims to stop caching the tokens that have been cached in the previous layers, can reduce the gap between the important and unimportant tokens, and prevent the background tokens from overlarge caching frequency.This observation has also been verified in Figure 9(c) that $s_3$ reduces the number of tokens cached by more than 4.5k times.

## 5 CONCLUSION

Motivated by the observation that different tokens exhibit different temporal redundancy and different error propagation, this paper introduces ToCa, a token-wise feature caching method, which adaptively skips the computation of some tokens by resuing their features in previous timesteps. By leveraging the difference in different tokens, ToCa achieves better acceleration performance compared with previous caching methods by a clear margin in both image and video generation, providing insights for token-wise optimization in diffusion transformers.

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

# A  APPENDIX

## A.1  ENGINEERING DETAILS

This section introduces some engineering techniques in our work.

### A.1.1  STEP-WISE CACHING SCHEDULING

In section 3.6, we propose a method for dynamically adjusting the caching ratio $R$ based on the time redundancy and noise diffusion speed across different depths and types of layers, which constitutes a key part of our contributions. In the following section, we further explore the dynamic adjustment of $R$ along the timestep dimension, as well as strategies for dynamically adjusting the forced activation cycle $\mathcal{N}$.

At the initial stages of image generation, the model primarily focuses on generating contours, while in the later stages, it pays more attention to details. In the early contour generation phase, it is not necessary for too many tokens to be fully computed with high precision. By multiplying by a term $r_t$, we achieve dynamic adjustment of $R$ along the timestep dimension, where $r_t = 0.5 + \lambda_t(0.5 - t/T)$, $\lambda_t$ is a positive parameter controlling the slope, $t$ is the number of timesteps already processed, and $T$ is the total number of timesteps. By adjusting $R$ in this way, we shift some of the computational load from earlier timesteps to later ones, improving the quality of the generated images. Finally, the caching ratio is determined as $R_{\text{type}}^{l,t} = R \times r_l \times r_{\text{type}} \times r_t$.

Similarly, we set a larger forced activation cycle $\mathcal{N}$ during the earlier stages, while a smaller $\mathcal{N}$ is used during the later detail generation phase to enhance the quality of the details. To ensure that the adjustment of $\mathcal{N}$ has minimal impact on the theoretical speedup, we define it as follows: $\mathcal{N}_t = \mathcal{N}_0/(0.5 + w_t(t/T - 0.5))$, where $\mathcal{N}_0$ corresponds to the expected theoretical speedup induced by $\mathcal{N}$, and $w_t$ is a hyperparameter controlling the slope.

### A.1.2  PARTIALLY COMPUTATION ON SELF-ATTENTION

In the previous section, we mentioned that partial computation in the Self-Attention module can lead to rapid propagation and accumulation of errors. Therefore, we considered avoiding partial computation in the Self-Attention module, meaning that during the non-forced activation phase, the Self-Attention module has $r_{type} = 0$. In the subsequent Sensitivity Study, we explored a trade-off scheme between Self-Attention and MLP modules, with the corresponding formulas for allocation

being $r_{type} = 1 - 0.4\lambda_{type}$ for the Self-Attention module, and $r_{type} = 1 + 0.6\lambda_{type}$ for the MLP module. The factors 0.6 and 0.4 are derived from the approximate computational ratio between these two modules in the DiT model.

### A.1.3 TOKEN SELECTION FOR CFG AND NON-CFG

In the series of DiT-based models, the tensors of cfg (class-free guidance) and non-cfg are concatenated along the batch dimension. A pertinent question in token selection is whether the same token selection strategy should be applied to both the cfg and non-cfg parts for the same image (i.e., if a token is cached in the cfg part, it should also be cached in the corresponding non-cfg part). We have observed significant sensitivity differences among models with different types of conditioning regarding whether the same selection strategy is used. For instance, in the text-to-image and text-to-video models, such as PixArt-$\alpha$ and OpenSora, if independent selection schemes are applied for the cfg and non-cfg parts, the model performance degrades substantially. Thus, it is necessary to enforce a consistent token selection scheme between the cfg and non-cfg parts.

However, in the class-to-image DiT model, this sensitivity issue is considerably reduced. Using independent or identical schemes for the cfg and non-cfg parts results in only minor differences. This can be attributed to the fact that, in text-conditional models, the cross-attention module injects the conditioning information into the cfg and non-cfg parts unevenly, leading to a significant disparity in attention distribution between the two. Conversely, in class-conditional models, the influence on both parts is relatively uniform, causing no noticeable changes in token attention distribution.

### A.2 MORE IMPLEMENTATION DETAILS ON EXPERIMENTAL SETTINGS

For the DiT-XL/2 model, we uniformly sampled from 1,000 classes in ImageNet (Deng et al., 2009) and generated 50,000 images with a resolution of $256 \times 256$. We explored the optimal solution for DiT-XL/2 using FID-5k (Heusel et al., 2017) and evaluated its performance with FID-50k. Additionally, sFID, Inception Score, and Precision and Recall were used as secondary metrics. For the PixArt-$\alpha$ model, we used 30,000 captions randomly selected from COCO-2017 (Lin et al., 2014) to generate 30,000 images. We computed FID-30k to assess image quality and used the CLIP Score between the images and prompts to evaluate the alignment between image content and the prompts. For the OpenSora model, we used the VBench framework (Huang et al., 2024), generating 5 videos for each of the 950 VBench benchmark prompts under different random seeds, resulting in a total of 4,750 videos. These videos have a resolution of 480p, an aspect ratio of 16:9, a duration of 2 seconds, and consist of 51 frames saved at a frame rate of 24 frames per second. The model was comprehensively evaluated across 16 aspects: subject consistency, imaging quality, background consistency, motion smoothness, overall consistency, human action, multiple objects, spatial relationships, object class, color, aesthetic quality, appearance style, temporal flickering, scene, temporal style, and dynamic degree.

**PixArt-$\alpha$**: We set the average forced activation cycle of `ToCa` to $\mathcal{N} = 2$, supplemented with a dynamic adjustment parameter $w_t = 0.1$. The parameter $\lambda_t = 0.4$ adjusts $R$ at different time steps, and the average caching ratio is $R = 70\%$. The parameter $r_l = 0.3$ adjusts $R$ at different depth layers. The module preference weight $r_{type} = 1.0$ shifts part of the computation from cross-attention layers to MLP layers.

**OpenSora**: For OpenSora, we fixed the forced activation cycle for temporal attention, spatial attention, and MLP at 3, and set the forced activation cycle for cross-attention to 6. The `ToCa` strategy ensures that a portion of token computations is conducted solely in the MLP, with $R_{mlp}$ fixed at 85%.

**DiT**: We set the average forced activation cycle of `ToCa` to $\mathcal{N} = 3$, supplemented with a dynamic adjustment parameter $w_t = 0.03$ to gradually increase the density of forced activations as the sampling steps progress. The parameter $\lambda_t = 0.03$ adjusts $R$ at different time steps. Additionally, during the sampling steps in the interval $t \in [50, 100]$, the forced activation cycle is fixed at $\mathcal{N} = 2$ to promote more thorough computation in sensitive regions. The average caching ratio is $R = 93\%$, and the parameter $\lambda_l = 0.06$ adjusts $R$ at different depth layers. The module preference weight $r_{type} = 0.8$ means that during steps outside the forced activation ones, no extra computations are performed in attention layers, but additional computations are performed in the MLP layers.

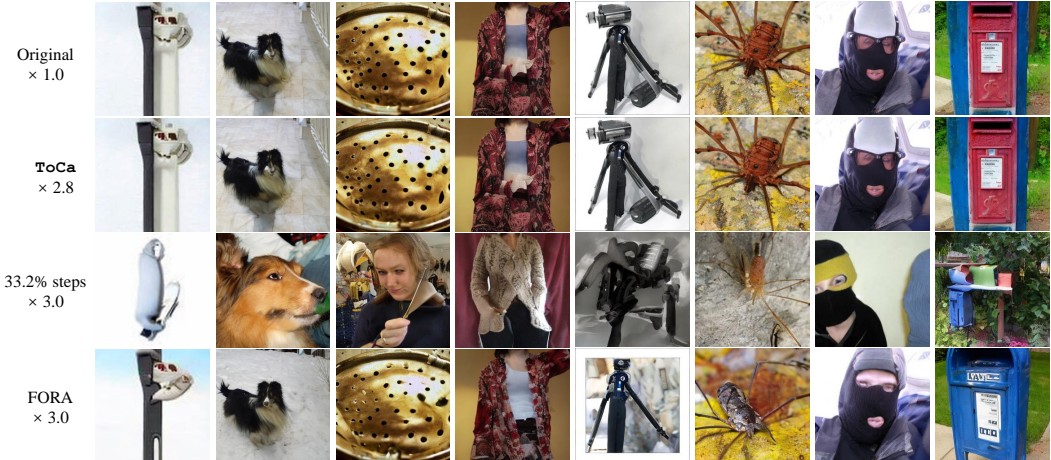

Figure 10: Visualization examples for different acceleration methods on DiT.

All of our experiments were conducted on 6 A800 GPUs, each with 80GB of memory, running CUDA version 12.1. The DiT model was executed in Python 3.12 with PyTorch version 2.4.0, while PixArt-$\alpha$ and OpenSora were run in Python 3.9. The PyTorch version for PixArt-$\alpha$ was 2.4.0, and for OpenSora it was 2.2.2. The CPUs used across all experiments were 84 vCPUs from an Intel(R) Xeon(R) Gold 6348 CPU @ 2.60GHz.

## A.3 SENSITIVITY STUDY

We explored the optimal parameter configuration for the `ToCa` acceleration scheme on DiT and analyzed the sensitivity of each parameter. The experiments used FID-5k and sFID-5k as evaluation metrics. From Figure 11 (a) to (f), we respectively investigated the effects of the caching ratio weights $\lambda_l$, $\lambda_{type}$, $\lambda_t$, the weight of Cache Frequency score $\lambda_3$, Uniform Spatial Distribution $\lambda_4$, and the dynamic adjustment weight for forced activation $w_t$. It is observed that: (a) The optimal parameter is $\lambda_l = 0.06$, where the corresponding cache ratio shows approximately $6\%$ variation at both the last and first layers. (b) The optimal parameter is $\lambda_{type} = 2.5$, at which point the Self-Attention layer does not perform any partial computation, with the entire computational load shifted to the MLP layer. It is also noted that as the computation load decreases in the Self-Attention layer and increases in the MLP layer, the generation quality shows a steady improvement. (c) The optimal parameter in the figure is $\lambda_t = 0.03$, and at this point, there is little difference between the best and worst methods, suggesting that the model is not particularly sensitive to the adjustment of cache ratio along the timesteps. (d) The optimal weight for the Cache Frequency score is $\lambda_3 = 0.25$. We observe that as $\lambda_3$ increases, the model's generation quality initially shows a noticeable improvement, but beyond a weight value of 0.25, the fluctuation is minimal. This indicates that the Cache Frequency has reached a relatively uniform state, achieving a dynamic balance in caching among different tokens. (e) We conducted a search for the Uniform Spatial Distribution score with grid sizes of 2 and 4, and the experimental results show that the generation quality with a grid size of 2 is generally better than that with a grid size of 4. This suggests that a finer-grained spatial uniformity indeed contributes to an improvement in generation quality. (f) We explored the impact of dynamically adjusting the forced activation cycle on the model's generation quality and analyzed the effect of fixing the Force activation cycle $\beta$ at 2 for the relatively more sensitive 50–100 timesteps. The experimental results show that enforcing this fixed cycle in the 50–100 timesteps significantly improves generation quality, with the optimal parameter configuration being $w_t = 0.4$.

In summary, these observations indicate that our method is not sensitive to the choice of hyper-parameters. Actually, our experiment results demonstrate that stable performance can be observed when directly transferring hyper-parameters from one model to another model in the same model family such as DiT in different sizes.

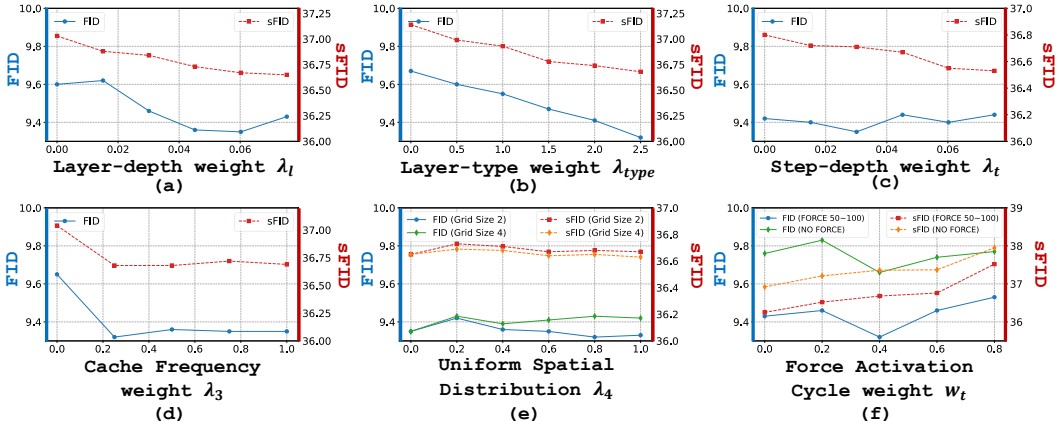

Figure 11: **Sensitivity study on different weights**. From (a) to (f), the caching ratio weights $\lambda_l$, $\lambda_{type}$, $\lambda_t$, the Cache Frequency score weight $\lambda_3$, the Uniform Spatial Distribution weight $\lambda_4$, and the dynamic schedule weight for forced activation $w_t$ are presented.

## A.4 COMPUTATION COMPLEXITY ANALYSIS

### A.4.1 MAIN COMPUTATIONS

**Complexity of Attention Layer.** In the Attention layer, tokens are first processed through a linear layer to generate queries, keys, and values. Next, the queries and keys are multiplied using a dot product, passed through a softmax function, and then multiplied with the values. Finally, the result is passed through another linear layer to produce the output. The computational cost of the Self-Attention layer can be expressed as:

$$
\begin{aligned}
\text{FLOPs}_{SA} \approx & N \times D \times 3 \times D \times 2 + N^2 \times D \times 2 + N^2 \times H \times 5 + N^2 \times D \times 2 + N \times D \times D \times 2 \\
= & 8 \times N \times D^2 + 4 \times N^2 \times D + 5 \times N^2 \times H \approx O(ND^2) + O(N^2 D),
\end{aligned}
\tag{4}
$$

where $N$ is the number of tokens, $D$ is the hidden state of each token and $H$ is the number of heads, $(H \ll D)$. The computational cost of a Cross-Attention layer can be expressed as:

$$
\begin{aligned}
\text{FLOPs}_{CA} \approx & N_1 \times D \times D \times 2 + N_2 \times D \times 2 \times D \times 2 + N_1 \times N_2 \times D \times 2 \\
& + N_1 \times N_2 \times H \times 5 + N_1 \times N_2 \times D \times 2 + N_1 \times D \times D \times 2 \\
= & 4 \times (N_1 + N_2) \times D^2 + 4 \times N_1 \times N_2 \times D + 5 \times N_1 \times N_2 \times H \\
\approx & O((N_1 + N_2)D^2) + O(N_1 N_2 D) = O((N + N_2)D^2) + O(NN_2 D),
\end{aligned}
\tag{5}
$$

where $N_1 = N, N_2$ are the number of image and text tokens, D is the hidden state of each token and H is the number of heads, $(H \ll D)$. In the previous computations, the softmax operation is approximated as involving 5 floating-point calculations per element.

**Complexity of MLP Layer.** The computational cost of MLP layer can be written as:

$$
\begin{aligned}
\text{FLOPs}_{MLP} \approx & N \times D_1 \times D_2 \times 2 + N \times D_2 \times 6 + N \times D_1 \times D_2 \times 2 \\
= & 4 \times N \times D_1 \times D_2 + 6 \times D_2 \times N \\
= & 16 \times N \times D^2 + 24 \times N \times D \approx O(ND^2),
\end{aligned}
\tag{6}
$$

where $N$ is the number of tokens, $D_1 = D$ and $D_2 = 4D_1$ are the hidden and middle-hidden state in MLP, respectively. The activation function for MLP is approximated to involve 6 floating-point operations per element.

### A.4.2 COMPUTATION COSTS FROM TOKEN SELECTION

**Self-Attention score $s_1$.** As mentioned in section 3.5, the Self-Attention score $s_1$ is computed as $s_1(x_i) = \sum_{i=1}^{N} \alpha_{ij}$, where $x_i$ is the $i_{th}$ token, and $\alpha_{ij}$ is the element in the self-attention map.

Table 6: **Quantitative comparison on class-to-image generation** on ImageNet with **50 steps DDIM sampler** as baseline on DiT-XL/2.

| Method | Latency(s) ↓ | FLOPs(T) ↓ | Speed ↑ | sFID ↓ | FID ↓ | Precision ↑ | Recall ↑ |
|---|---|---|---|---|---|---|---|
| **DiT-XL/2-G (cfg = 1.50)** | 0.455 | 23.74 | 1.00× | 4.40 | 2.43 | 0.80 | 0.59 |
| 50% **steps** | 0.238 | 11.86 | 2.00× | 4.74 | 3.18 | 0.79 | **0.58** |
| 40% **steps** | 0.197 | 9.50 | 2.50× | 5.15 | 3.81 | 0.78 | 0.57 |
| 34% **steps** | 0.173 | 8.08 | 2.94× | 5.76 | 4.58 | 0.77 | 0.56 |
| **FORA**($\mathcal{N} = 2.5$) | 0.219 | 10.48 | 2.27× | 6.59 | 3.83 | 0.79 | 0.55 |
| **FORA**($\mathcal{N} = 3$) | 0.211 | 8.58 | 2.77× | 6.43 | 3.88 | 0.79 | 0.54 |
| `ToCa` ($\mathcal{N} = 3, R = 93\%$) | 0.227 | 10.23 | 2.32× | **4.74** | **3.04** | **0.80** | 0.57 |
| `ToCa` ($\mathcal{N} = 4, R = 93\%$) | 0.209 | 8.73 | 2.72× | 5.11 | 3.60 | 0.79 | 0.56 |

Therefore, the computational complexity of the self-attention score is only $N \approx O(N)$. In a practical case achieving about $2.3\times$ acceleration, the computation cost of the Self-Attention score accounts for approximately $0.28\%$ of the main components.

**Cross-Attention score $s_2$.** As mentioned in section 3.5, the Cross-Attention score $s_2$ is computed as $s_2(x_i) = -\sum_{j=1}^{N} c_{ij} \log(c_{ij})$, where the $c_{ij}$ is the element in the cross-attention map. Therefore, the computational complexity of the cross-attention score is only $2N \approx O(N)$. In a practical case achieving about $2.3\times$ acceleration, the computation cost of the Cross-Attention score accounts for approximately $0.35\%$ of the main components.

**Cache Frequency score $s_3$ and Uniform Spatial Distribution score $s_4$.** The Cache Frequency score $s_3$ is updated at each step, so its update cost per timestep is $N$. When the Cache Frequency score is called for summation in practical applications, the computation cost is $2N$. Thus, the total cost for one layer is $3N \approx O(N)$. The Uniform Spatial Distribution score $s_4$ is computed by sorting within each grid of size $G \times G$ and weighting the top-scoring tokens. The computation cost is given by $\frac{N}{G^2} \times G^2 \log(G^2) + 2 \times \frac{N}{G^2}$, where $G$ is the grid size, which is usually small. Therefore the computational complexity of $s_4$ is $O(N)$. In a practical case achieving about $2.3\times$ acceleration, the computation cost of the Cache Frequency score $s_3$ accounts for approximately $0.044\%$ and the Uniform Spatial Distribution score $s_4$ accounts for $0.15\%$ of the main computation components. In addition, the computational cost for sorting $N$ tokens is $O(N \log N)$, which accounts for approximately$0.18\%$ of the main computational cost.

In summary, although `ToCa` introduces additional computations, its computational complexity of $O(N)$ is negligible compared to the main computational modules with complexities of $O(N^2 D)$ or $O(ND^2)$. In practical tests, the time taken for token selection is minimal, typically less than $1\%$ of the main computational cost. At cache steps, taking a caching ratio of $\mathcal{R} = 90\%$ as an example, the computational cost of terms with a complexity of $O(ND^2)$ is reduced to $10\%$ of the original, while the computational cost of terms with a complexity of $O(N^2 D)$ is reduced to $1\%$. (However, as mentioned earlier, in practice, it is more efficient to shift all computations at cache steps to the MLP. Therefore, all terms with a complexity of $O(N^2 D)$ at cache steps are ignored.)

### A.5 IMPLEMENTED RESULTS ON CLASS-CONDITIONAL IMAGE GENERATION

In addition to the series of experiments conducted using the DDPM(Ho et al., 2020) sampling method on DiT, which have already been included in the main part, we also performed validation with the more practically relevant DDIM(Song et al., 2021) sampling method to further demonstrate the effectiveness of `ToCa` as shown in Table 6.

For instance, `ToCa` leads to 1.32 and 0.28 lower values in sFID and FID compared with FORA with a similar acceleration ratio of approximately $2.7\times$, respectively, and achieves 0.21 lower values in FID compared to the method of directly reducing the sampling steps with an acceleration ratio of approximately $2.5\times$. As a trade-off between acceleration and performance, we selected the scheme $\mathcal{N} = 3, R = 93\%$ as the final recommended approach for DDIM sampler.

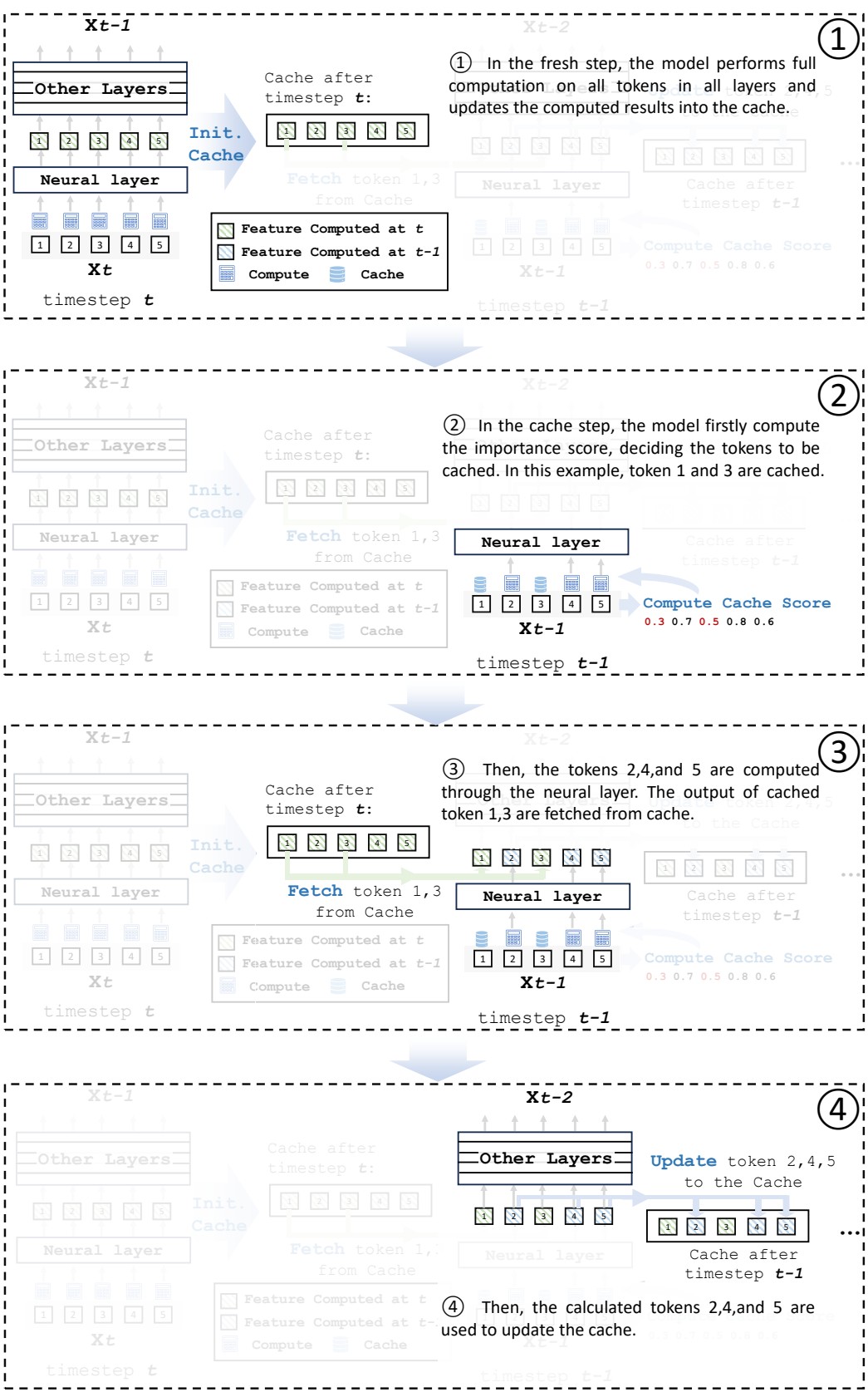

Figure 12: A more detailed workflow for the proposed ToCa.The cache-and-reuse procedure is conducted on the model at all layers and timesteps.

Table 7: **Quantitative comparison in text-to-image generation** for FLUX on Image Reward.

| Method | Latency(s) ↓ | FLOPs(T) ↓ | Speed ↑ | Image Reward ↑ |
|---|---|---|---|---|
| **FLUX.1-dev** (Labs, 2024) | 33.85 | 3719.50 | 1.00× | 1.202 |
| 68% **steps** | 23.02 | 2529.26 | 1.47× | 1.200 |
| **FORA** (Selvaraju et al., 2024) | 20.82 | 2483.32 | 1.51× | 1.196 |
| **ToCa**($\mathcal{N} = 2, R = 90\%$) | **19.88** | **2458.06** | **1.51×** | **1.202** |
| **FLUX.1-schnell** (Labs, 2024) | 2.882 | 277.88 | 1.00× | 1.133 |
| 75% **steps** | 2.162 | 208.41 | 1.33× | 1.139 |
| **FORA**[1] (Selvaraju et al., 2024) | 2.365 | 225.60 | 1.23× | 1.129 |
| **FORA**[2] (Selvaraju et al., 2024) | 2.365 | 225.60 | 1.23× | 1.124 |
| **FORA**[3] (Selvaraju et al., 2024) | 2.365 | 225.60 | 1.23× | 1.123 |
| **ToCa**($\mathcal{N} = 2, R = 90\%$) | **1.890** | **181.30** | **1.53×** | **1.134** |

## A.6 RESULTS ON HIGHER-RESOLUTION AND MORE ADVANCED TEXT-TO-IMAGE MODELS

As shown in Table 7, we compared the performance of FORA and ToCa in generating high-resolution images ($1024 \times 1024$) using the more advanced text-conditional image generation models FLUX.1-dev and FLUX.1-schnell(Labs, 2024). The former uses 50 sampling steps, while the latter, as a more efficient model, uses only 4 sampling steps. The evaluation of generation quality was conducted using Image Reward, a metric better suited to measuring human preference. The generated images were based on 1,632 prompts from the PartiPrompts(Yu et al., 2022) dataset to comprehensively evaluate the generation quality of the acceleration methods on both FLUX.1-dev and FLUX.1-schnell.

In this comparison, FORA (Selvaraju et al., 2024) represents acceleration on the FLUX.1-dev model with 50 sampling steps, where caching is performed every other step (i.e., $\mathcal{N} = 2$). FORA[1], FORA[2], and FORA[3] correspond to skipping the 2nd, 3rd, and 4th steps, respectively, during the 4-step generation process. ToCa demonstrated nearly lossless performance under a $1.5\times$ acceleration, with Image Reward scores almost identical to the non-accelerated scenario in both the 50-step FLUX.1-dev and the 4-step FLUX.1-schnell models. In contrast, all configurations of FORA showed quality degradation. For example, in the 50-step FLUX.1-dev model, the $1.5\times$ acceleration setting and in the 4-step FLUX.1-schnell model, a $1.2\times$ acceleration setting both resulted in noticeable quality degradation. The corresponding visual results for FLUX.1-schnell are presented in Figure 13, demonstrating the lossless acceleration capability of ToCa.

## A.7 DETAILS FOR DISTRIBUTION FIGURES

In this section, we provide detailed explanations of the various distribution plots mentioned in the main text as supplementary information.

Figure 1 illustrates the distribution of the Frobenius norm of the differences between the feature maps $x_t^L$ and $x_{t+1}^L$, which are the outputs of the last DiT Block at each timestep $t$ and the previous timestep $t + 1$, respectively. The figure also presents the corresponding statistical frequency density of these Frobenius norm values for each token, based on 500 randomly generated samples produced by DiT. This analysis reveals the conclusion that different tokens exhibit varying levels of temporal redundancy.

Figure 2 illustrates the varying rates of error accumulation and propagation across different tokens. Specifically, Gaussian noise with an intensity of 0.5 was independently added to the $i_{th}$ token of the first layer at each step. The Frobenius norm was then computed between the output features of all tokens at the last layer of the same step and the corresponding features from the noise-free output. This process was repeated for all steps and all layers. Given that each noise propagation required re-running the inference process, a random subset of 100 samples from the DiT model was selected for this case study, and the noise propagation results were recorded for each iteration. This analysis led to the conclusion that different tokens exhibit varying rates of error accumulation and propagation.

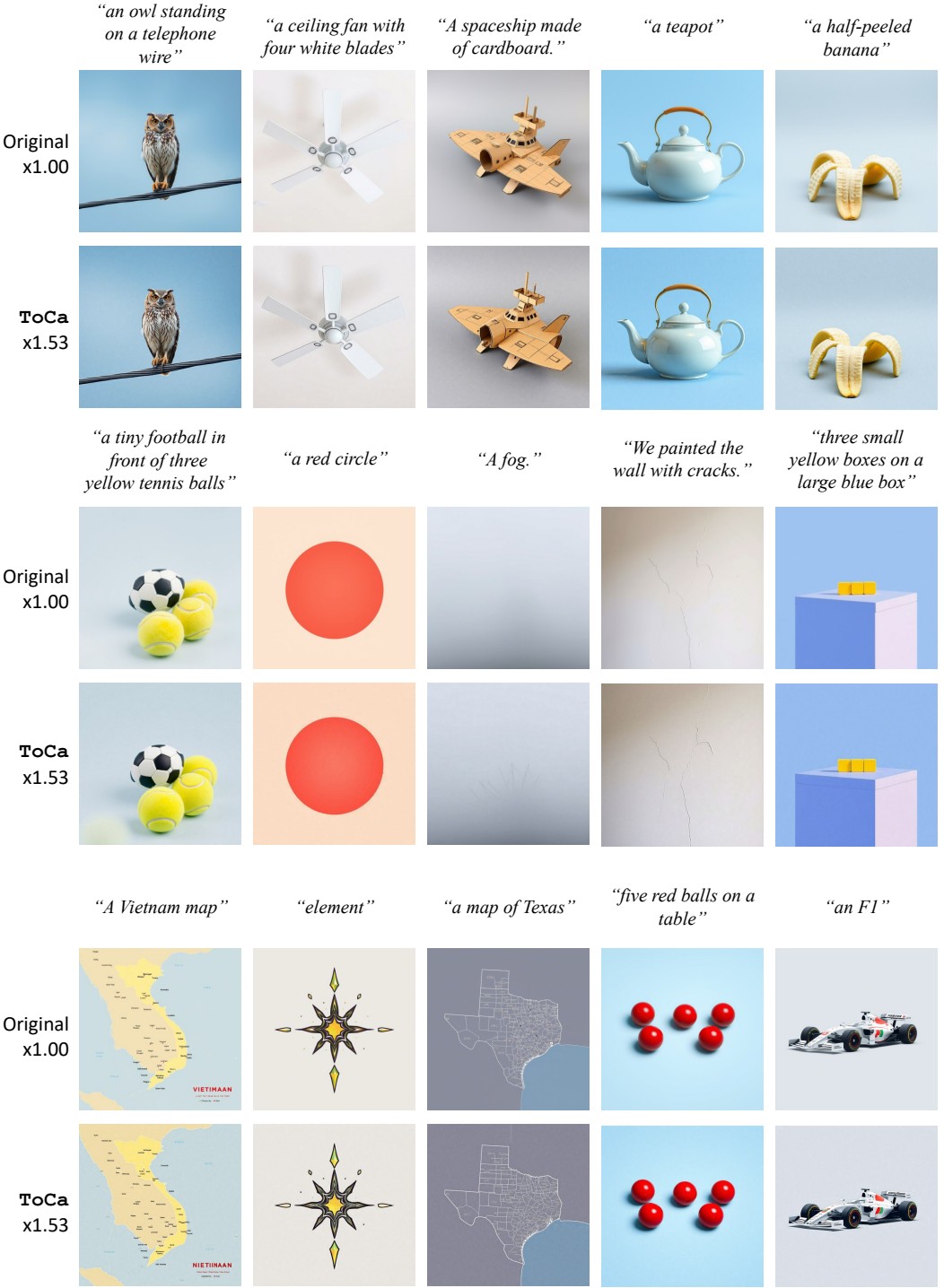

Figure 13: Visualization examples for original FLUX.schnell(Labs, 2024) and proposed `ToCa` with almost lossless acceleration.

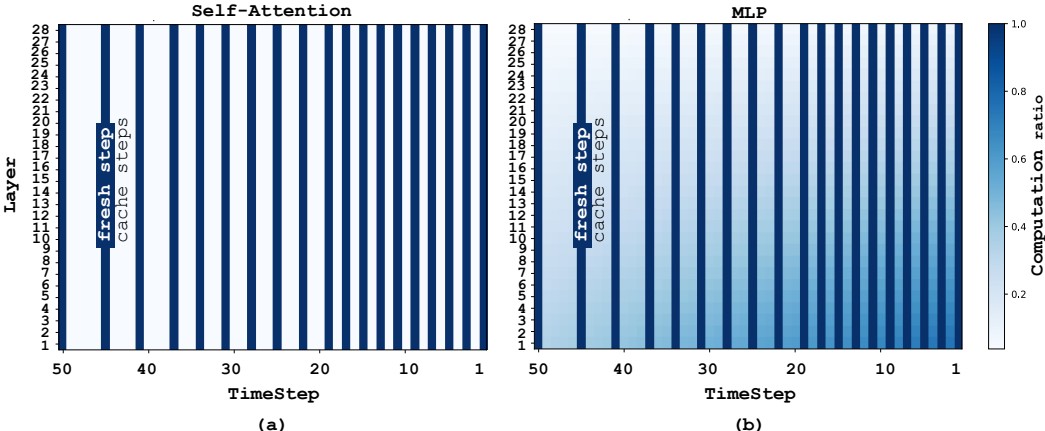

Figure 14: Computation ratio for different types of computation layers on different timesteps and layers on DiT. The dark blue lines correspond to fully computed fresh steps. (a) The computation ratio distribution of the Self-Attention layers. As mentioned earlier, performing partial computation on attention layers is less cost-effective compared to MLP layers. Therefore, we do not implement partial computation on attention layers; apart from fresh steps, all other steps directly reuse the corresponding cached features. (b) The computation ratio distribution of the MLP layers. As shown, the computation ratio increases with deeper layers and as the number of inferred timesteps during the inference phase grows.

Figure 5(a) illustrates the varying temporal redundancy across layers of different depths. For each timestep, the Frobenius norm of the differences between the features of the current timestep and the corresponding features of the previous timestep at a specific layer depth was computed for each token. The resulting Frobenius norm values were then used to plot the distribution alongside their corresponding statistical frequency densities. To clearly demonstrate the trends, we selected layers 1, 15, and 28 for visualization. The samples used were randomly chosen from 200 DiT samples.

Figure 5(b) shows the variation in the offset distribution of the output values from the last layer of a timestep when Gaussian noise is added to a single token at different layer depths within the same timestep. This is measured by adding Gaussian noise with an intensity of 0.5 to a single token in a specific layer at one timestep and comparing the deviation in the output features of the last layer at that timestep with the noise-free scenario. For clarity, this operation was performed on layers 1, 15, and 25 across all timesteps, using 200 randomly selected samples from the DiT model to generate the examples. It is worth noting that Figure 5(b) may appear at first glance to violate the normalization condition for frequency density distributions. This is due to the large variations in Frobenius norm values, which necessitated the use of a logarithmic scale on the horizontal axis. Figure 5(a) and (b) demonstrate two key conclusions: deeper layers exhibit poorer temporal redundancy, but errors introduced in deeper layers have a smaller impact on the output at the same timestep.

Figure 5(c) illustrates the results on the PixArt-$\alpha$ model by adding Gaussian noise with an intensity of $0.5 \times \|x_k\|_F$ to a single token $x_k$ in the 10th layer (approximately the middle layer) at each timestep. This process was performed for three different types of layers (self-attention, cross-attention, and MLP). The Frobenius norm of the error induced by the noise was measured on the output of the last layer and normalized by the average Frobenius norm $\|x_k\|_F$ of tokens of the same type. The resulting distribution was plotted using 200 prompts randomly selected from the MS-COCO2017 dataset. It is important to note that the additional normalization step, based on the norm values of the tokens, was necessary because the norm values of tokens in self-attention, cross-attention, and MLP layers typically vary significantly. Normalization ensures a fair comparison across these layer types. Additionally, the distribution for the MLP layer in Figure 5(c) appears more dispersed. This is due to the generally larger variations in MLP output values across different prompts and timesteps. In practice, increasing the number of samples can make the distribution visually denser. However, given that each token requires a separate inference for the error propagation experiments, using 200 prompts already incurs a significant computational cost while remaining sufficient to reveal the trends.

---

**Algorithm 1** `ToCa`

---

**Input:** current timestep $t$, current layer id $l$.

1: **if** current timestep $t$ is a fresh step **then**
2:     Fully compute $\mathcal{F}^l(x)$.
3:     $\mathcal{C}^l(x) := \mathcal{F}^l(x)$; # *Update the cache.*
4: **else**
5:     $\mathcal{S}(x_i) = \sum_{j=1}^{4} \lambda_j \cdot s_j$; # *Compute the cache score for each token.*
6:     $\mathcal{I}_{Compute} := \text{TopK}(\mathcal{S}(x_i), R\%)$; # *Fetch the index of computed tokens.*
7:     **for all** tokens $x_i$ **do**
8:         **if** $i \in \mathcal{I}_{Compute}$ **then**
9:             Compute $\mathcal{F}^l(x_i)$ through the neural layer.
10:            $\mathcal{C}^l(x_i) := \mathcal{F}^l(x_i)$; # *Update the cache.*
11:        **end if**
12:    **end for**
13: **end if**
14: **return** $\mathcal{F}^l(x)$. # *return features for both cached and computed tokens for the next layer.*

---

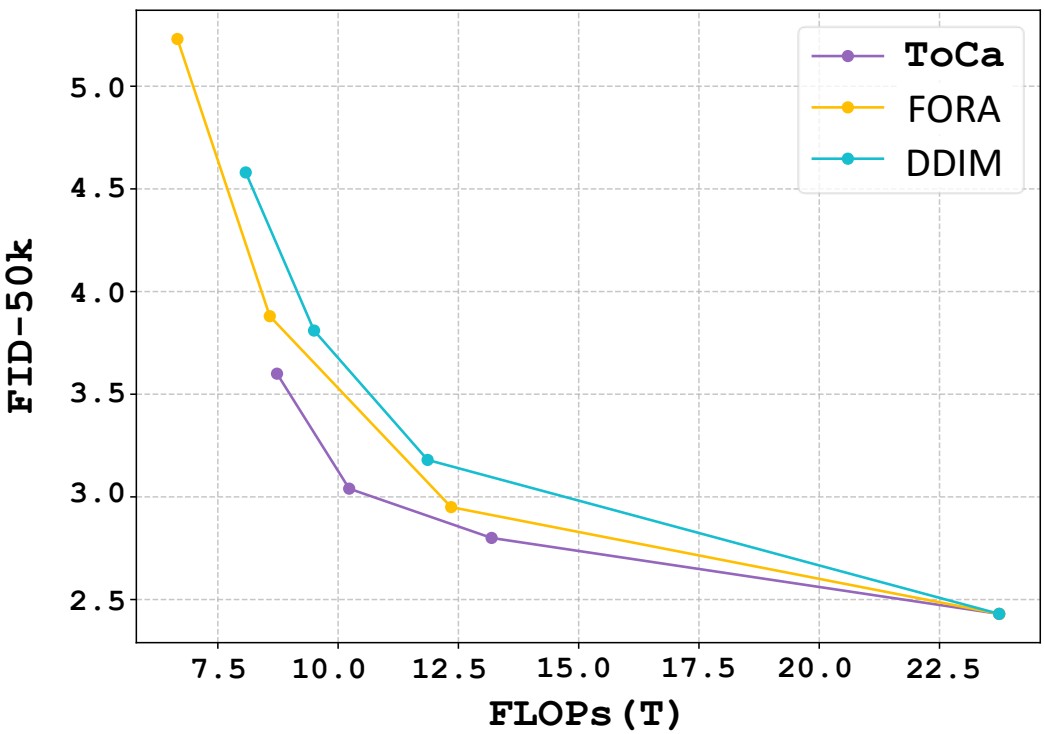

Figure 15: Pareto curve with FLOPs-FID to better evaluate the performance of `ToCa` on DiT with 50 ddim sampling steps as baseline.

