# OpenReview forum: "Accelerating Diffusion Transformers with Token-wise Feature Caching"
_ICLR.cc/2025/Conference — ICLR 2025 Poster_

### Official Review · Reviewer_R8q5 · 2024-10-29

**Soundness:** 3
**Presentation:** 3
**Contribution:** 2
**Rating:** 6
**Confidence:** 4

**Summary:**

The authors aim to accelerate DiT inference by caching features and reusing them in future steps. The authors observe the limitations of existing methods that ignore the sensitivities to feature caching among different tokens, thus leading to potential degradation of generation quality. Based on this observation, the main idea of this work is to selectively cache suitable tokens and adjust the caching ratios by considering layer types and depths. Evaluations based on three typical DiT models show the performance of model accuracy and overhead saving.

**Strengths:**

1.	The author provides two insightful observations that improve my understanding of diffusion transformers. Firstly, different tokens exhibit varying levels of temporal redundancy across different timesteps. Secondly, the authors highlight that due to error propagation in transformer architectures, the same caching error can lead to vastly different impacts on the final generation result. These observations are the basis of the proposed token-wise feature caching method.
2.	Unlike previous works that primarily focused on U-Net architectures, the authors shift their attention to the transformer architecture. This is meaningful as existing methods have not fully exploited the unique properties of transformers, which are increasingly popular for visual generation tasks.
3.	The experimental results demonstrate the performance improvements with high inference speedups.
4.	The authors select the most suitable tokens for caching by leveraging both temporal redundancy and error propagation. The proposed method adaptively caches the tokens that minimize the resulting error while maximizing the acceleration ratio.

**Weaknesses:**

1.	In Figure 1, the authors use the Frobenius distance to measure the temporal redundancy of tokens across timesteps. However, it is unclear how this distance is exactly calculated. Providing more details on the calculation process would help readers better understand the experiments.
2.	I cannot find a clear formulation for the cache score used in token selection. The authors mention the cache score in Figure 4, but no explicit equation is provided.
3.	The authors update the cache at all timesteps to reduce the error introduced by feature reusing. However, it is unclear how this procedure affects the computation complexity, especially about I/O overhead and cache selection efficiency.
4.	The experiments report the inference speedups but lack a comprehensive analysis of the trade-off between speedups and model accuracy. While FID and CLIP scores are provided in Table 1, these metrics do not fully reflect model accuracy. In addition, Table 3 shows that the proposed method has no significant improvements over existing methods in terms of accuracy and speedups.

**Questions:**

Please refer to the questions in the weakness part.

---

> ### Author Response · Authors · 2024-11-24
> **Response to Reviewer R8q5 Part 1/4**
>
> **W1: In Figure 1, the authors use the Frobenius distance to measure the temporal redundancy of tokens across timesteps. However, it is unclear how this distance is exactly calculated. Providing more details on the calculation process would help readers better understand the experiments.**
>
> **A1:** Thank you for your suggestion. Further detailing the distribution figures we have plotted will indeed help readers better understand the underlying principles of these distributions. In the updated PDF version, we have added a more detailed explanation of Figures 1, 2, 5(a), 5(b), and 5(c) in **Appendix A.7 DETAILS FOR DISTRIBUTION FIGURES**.
>
> Figure 1 illustrates the distribution of the Frobenius norm of the differences between the feature maps $x^L_t$ and $x^L_{t+1}$, which are the outputs of the last DiT Block at each timestep $t$ and the previous timestep $t+1$, respectively. The figure also presents the corresponding statistical frequency density of these Frobenius norm values for each token, based on 500 randomly generated samples produced by DiT. This analysis reveals the conclusion that different tokens exhibit varying levels of temporal redundancy.
>
> Figure 2 illustrates the varying rates of error accumulation and propagation across different tokens. Specifically, Gaussian noise with an intensity of 0.5 was independently added to the $i_{th}$ token of the first layer at each step. The Frobenius norm was then computed between the output features of all tokens at the last layer of the same step and the corresponding features from the noise-free output. This process was repeated for all steps and all layers. Given that each noise propagation required re-running the inference process, a random subset of 100 samples from the DiT model was utilized for this case study, and the noise propagation results were recorded for each iteration. This analysis led to the conclusion that different tokens exhibit varying rates of error accumulation and propagation.
>
> Figure 5 (a) illustrates the varying temporal redundancy across layers of different depths. For each timestep, the Frobenius norm of the differences between the features of the current timestep and the corresponding features of the previous timestep at a specific layer depth was computed for each token. The resulting Frobenius norm values were then used to plot the distribution alongside their corresponding statistical frequency densities. To clearly demonstrate the trends, we selected layers 1, 15, and 28 for visualization. The samples used were randomly chosen from 200 DiT samples.
>
> Figure 5 (b) shows the variation in the offset distribution of the output values from the last layer of a timestep when Gaussian noise is added to a single token at different layer depths within the same timestep. This is measured by adding Gaussian noise with an intensity of 0.5 to a single token in a specific layer at one timestep and comparing the deviation in the output features of the last layer at that timestep with the noise-free scenario. For clarity, this operation was performed on layers 1, 15, and 25 across all timesteps, using 200 randomly selected samples from the DiT model to generate the examples. It is worth noting that Figure 5 (b) may appear at first glance to violate the normalization condition for frequency density distributions. This is due to the large variations in Frobenius norm values, which necessitated the use of a logarithmic scale on the horizontal axis. Figure 5 (a) and (b) demonstrate two key conclusions: deeper layers exhibit poorer temporal redundancy, but errors introduced in deeper layers have a smaller impact on the output at the same timestep.
>
> Figure 5 (c) illustrates the results on the PixArt-$\alpha$ model by adding Gaussian noise with an intensity of $0.5 \times \|x_k\|_F$ to a single token $x_k$ in the 10th layer (approximately the middle layer) at each timestep. This process was performed for three different types of layers (self-attention, cross-attention, and MLP). The Frobenius norm of the error induced by the noise was measured on the output of the last layer and normalized by the average Frobenius norm $\|x_k\|_F$ of tokens of the same type. The resulting distribution was plotted using 200 prompts randomly selected from the MS-COCO2017 dataset.
>
> It is important to note that the additional normalization step, based on the norm values of the tokens, was necessary because the norm values of tokens in self-attention, cross-attention, and MLP layers typically vary significantly. Normalization ensures a fair comparison across these layer types. Additionally, the distribution for the MLP layer in Figure  appears more dispersed. This is due to the generally larger variations in MLP output values across different prompts and timesteps.
>
> Due to space constraints, we make the rest analysis in Response Part 2.

---

> ### Author Response · Authors · 2024-11-24
> **Response to Reviewer R8q5 Part 2/4**
>
> In practice, increasing the number of samples can make the distribution visually denser. However, given that each token requires a separate inference for the error propagation experiments, using 200 prompts already incurs a significant computational cost while remaining sufficient to reveal the trends.
>
> Generally, for figures such as Figure 1 and 5 (a) that consider temporal redundancy across tokens, the Frobenius norm of the differences between the output at each timestep at a specified position and the output at the same position in the previous timestep is computed for each token. This is used to plot distributions that describe temporal redundancy. For figures such as Figure 2 and 5 (b) and (c), noise is added to a single token, and the error propagation across multiple layers is measured in order to assess the distribution of error accumulation and propagation. We use Figure 2 as an example for explanation. Figure 2 shows that Gaussian noise with a fixed intensity of 0.5 is added to a token in the first DiT block at each timestep. The Frobenius norm of the difference between the noisy output and the noise-free output at the last layer is measured, representing the distance for this case.
>
> **W2: I cannot find a clear formulation for the cache score used in token selection. The authors mention the cache score in Figure 4, but no explicit equation is provided.**
>
> **A2:** Thank you for your suggestion. We acknowledge that the explanation of the cache score in our draft was not sufficiently clear. In the updated PDF version, we have explicitly provided the formulation for the cache score as follows:
>
> $\mathcal{S}(x_i) = \lambda_1 \sum_{j=1}^{N} \alpha_{ij} - \lambda_2 \sum_{j=1}^{N} c_{ij} \log(c_{ij}) + \lambda_3 \frac{n_{i}}{\mathcal{N}} + \lambda_4 \mathcal{I}(x_{i}) \cdot \left( \lambda_1 \sum_{j=1}^{N} \alpha_{ij} - \lambda_2 \sum_{j=1}^{N} c_{ij} \log(c_{ij}) + \lambda_3 \frac{n_{i}}{\mathcal{N}} \right)$,
>
> where $\lambda_k$ are the balance factors, $\alpha_{ij}$ is the element in the self-attention map, $c_{ij}$ is the element in the cross-attention map, $n_{i}$ denotes the cached times for token $x_i$, $\mathcal{N}$ is the number of timesteps in each feature caching cycle, $\mathcal{I}$ is the signal function, and $\mathcal{I}(x_{i})$ is an indicator function that equals 1 if $x_{i}$ has the highest score of $\lambda_1 \cdot s_1(x_i) + \lambda_2 \cdot s_2(x_i) + \lambda_3 \cdot s_3(x_i)$ in its neighboring $k \times k$ pixels and equals 0 otherwise.
>
> **W3: The authors update the cache at all timesteps to reduce the error introduced by feature reusing. However, it is unclear how this procedure affects the computation complexity, especially about I/O overhead and cache selection efficiency.**
>
> **A3:** Thank you for your suggestion. Conducting further computational analysis is indeed helpful for understanding the acceleration method of ToCa. In the updated version of the PDF, we provide a detailed breakdown of the computational costs of each major computation layer and the token-selection method in **Appendix A.4 COMPUTATION COMPLEXITY ANALYSIS**.
>
> In summary, the computation costs of self-attention score (s1), cross-attention score (s2), caching frequency score (s3), spatial bonus (s4) and token sorting **only account for 0.28%, 0.35%, 0.04%, 0.15% and 0.18% of the whole computation costs, respectively**.
>
> In the following Response Part 3, we introduce the computation complexity for attention and MLP layers, as well as the computation costs for token selection in detail.

---

> ### Author Response · Authors · 2024-11-24
> **Response to Reviewer R8q5 Part 3/4**
>
> **Attention layer.** In the Attention layer, tokens are first processed through a linear layer to generate queries, keys, and values. Next, the queries and keys are multiplied using a dot product, passed through a softmax function, and then multiplied with the values. Finally, the result is passed through another linear layer to produce the output.
>
> The computational cost of the Self-Attention layer can be expressed as:
>
> $\text{FLOPs}_{SA} \approx N \times D \times 3 \times D \times 2 + N^2 \times D \times 2 + N^2 \times H \times 5 + N^2 \times D \times 2 + N \times D \times D \times 2$
> $= 8 \times N \times D^2 + 4 \times N^2 \times D + 5 \times N^2 \times H \approx O(ND^2) + O(N^2D),$
>
> where $N$ is the number of tokens, $D$ is the hidden state of each token, and $H$ is the number of heads ($H \ll D$).
>
>
> The computational cost of a Cross-Attention layer can be expressed as:
>
> $\text{FLOPs}_{CA} \approx N_1 \times D \times D \times 2 + N_2 \times D \times 2 \times D \times 2 + N_1 \times N_2 \times D \times 2 + N_1 \times N_2 \times H \times 5 + N_1 \times N_2 \times D \times 2 + N_1 \times D \times D \times 2$
> $= 4 \times (N_1 + N_2) \times D^2 + 4 \times N_1 \times N_2 \times D + 5 \times N_1 \times N_2 \times H$
> $\approx O((N_1 + N_2)D^2) + O(N_1 N_2D) = O((N + N_2)D^2) + O(N N_2D),$
>
> where $N_1 = N$, $N_2$ are the number of image and text tokens, $D$ is the hidden state of each token, and $H$ is the number of heads ($H \ll D$). In the previous computations, the softmax operation is approximated as involving 5 floating-point calculations per element.
>
> **MLP layer.** The computational cost of the MLP layer can be written as:
>
> $\text{FLOPs}_{MLP} \approx N \times D_1 \times D_2 \times 2 + N \times D_2 \times 6 + N \times D_1 \times D_2 \times 2$
> $= 4 \times N \times D_1 \times D_2 + 6 \times D_2 \times N$
> $= 16 \times N \times D^2 + 24 \times N \times D \approx O(ND^2),$
>
> where $N$ is the number of tokens, $D_1 = D$, and $D_2 = 4D_1$ are the hidden and middle-hidden states in the MLP, respectively. The activation function for the MLP is approximated to involve 6 floating-point operations per element.
>
> Besides, there are computational costs by Token Selection process, as analyzed below:
>
> **Self-Attention Score $s_1$.** As mentioned in section 3.5, the Self-Attention score $s_1$ is computed as $s_1(x_i) = \sum_{i=1}^{N} \alpha_{ij}$, where $x_i$ is the $i_{th}$ token, and $\alpha_{ij}$ is the element in the self-attention map. Therefore, the computational complexity of the self-attention score is only $N \approx O(N)$. In a practical case achieving about $2.3\times$ acceleration, the computation cost of the Self-Attention score in TOCA accounts for approximately $0.28$% of the main components.
>
> **Cross-Attention Score $s_2$.** As mentioned in section 3.5 the Cross-Attention score $s_2$ is computed as $s_2(x_i) = -\sum_{j=1}^{N} c_{ij} \log (c_{ij})$,  where the $c_{ij}$ is the element in the cross-attention map. Therefore, the computational complexity of the cross-attention score is only $2N \approx O(N)$. In a practical case achieving about $2.3\times$ acceleration, the computation cost of the Cross-Attention score in TOCA accounts for approximately $0.35$% of the main components.
>
> **Cache Frequency Score $s_3$ and Uniform Spatial Distribution Score $s_4$.** The Cache Frequency score $s_3$ is updated at each step, so its update cost per timestep is $N$. When the Cache Frequency score is called for summation in practical applications, the computation cost is $2N$. Thus, the total cost for one layer is $3N \approx O(N)$. The Uniform Spatial Distribution score $s_4$ is computed by sorting within each grid of size $G \times G$ and weighting the top-scoring tokens. The computation cost is given by $\frac{N}{G^2} \times G^2 \log(G^2)  + 2 \times \frac{N}{G^2}$, where $G$ is the grid size, which is usually small. Therefore the computational complexity of $s_4$ is $O(N)$. In a practical case achieving about $2.3\times$ acceleration, the computation cost of the Cache Frequency score $s_3$ accounts for approximately $0.044$%  and the Uniform Spatial Distribution score $s_4$ accounts for $0.15$% of the main computation components. In addition, the computational cost for sorting $N$ tokens is $O(N\log N)$, which accounts for approximately $0.18$% of the main computational cost.
>
> Due to space constraints, we summary the additional computations in Response Part 4.

---

> ### Author Response · Authors · 2024-11-24
> **Response to Reviewer R8q5 Part 4/4**
>
> In summary, although ToCa introduces additional computations, its computational complexity of $O(N)$ is negligible compared to the main computational modules with complexities of $O(N^2 D)$ or $O(N D^2)$. In practical tests, the time taken for token selection is minimal, typically less than $1\%$ of the main computational cost. At cache steps, taking a caching ratio of $\mathcal{R}=90\%$ as an example, the computational cost of terms with a complexity of $O(N D^2)$ is reduced to $10\%$ of the original, while the computational cost of terms with a complexity of $O(N^2 D)$ is reduced to $1\%$. As mentioned earlier, in practice, it is more efficient to shift all computations at cache steps to the MLP. Therefore, all terms with a complexity of $O(N^2 D)$ at cache steps are ignored.
>
> In addition, regarding the I/O overhead of ToCa, for all cache steps in the self-attention layers of ToCa, the complete features are directly retrieved from the cache, eliminating the need to load the corresponding parameters. Furthermore, the memory access overhead for individual tokens in ToCa is negligible compared to the major computation layers.
>
> **W4: The experiments report the inference speedups but lack a comprehensive analysis of the trade-off between speedups and model accuracy. While FID and CLIP scores are provided in Table 1, these metrics do not fully reflect model accuracy. In addition, Table 3 shows that the proposed method has no significant improvements over existing methods in terms of accuracy and speedups.**
>
> **A4:** Thank you for your suggestion. In the updated PDF, we present in **Appendix Table 6** (Table 1 below) the evaluation of ToCa on the more advanced FLUX.1-schnell model [1] using the Image Reward metric, which better reflects human preferences. In this experiment, ToCa demonstrates a 1.5x lossless acceleration capability, outperforming FORA [2] under various parameter configurations.
>
> Additionally, we include in **Appendix Table 7** (Table 2 below) the acceleration results on the DiT model with 50 DDIM sampling steps as the baseline and provide a FLOPs-FID performance curve in **Figure 15** to illustrate the trade-off between acceleration and model precision. The additional experimental results should sufficiently demonstrate the superiority of ToCa compared to other approaches.
>
> **Table1: Quantitative comparison in text-to-image generation for FLUX on Image Reward. FORA$^{1}$, FORA$^{2}$, and FORA$^{3}$ correspond to skipping the 2nd, 3rd, and 4th steps, respectively, during the 4-step generation process.**
>
> | Method                          | Latency(s) ↓ | FLOPs(T) ↓ | Speed ↑ | Image Reward ↑ |
> |---------------------------------|--------------|------------|---------|----------------|
> | FLUX.1-schnell                  | 2.882        | 277.88     | 1.00×   | 1.133          |
> | FORA¹                           | 2.365        | 225.60     | 1.23×   | 1.129          |
> | FORA²                           | 2.365        | 225.60     | 1.23×   | 1.124          |
> | FORA³                           | 2.365        | 225.60     | 1.23×   | 1.123          |
> | ToCa ($\mathcal{N}=2,R=0.90$)   | **1.890**        | **181.30**    | **1.53×**   | **1.134**          |
>
> **Table2: Quantitative comparison on class-to-image generation on ImageNet with 50 steps DDIM sampler as baseline on DiT-XL/2.**
>
> | Method                       | Latency(s) ↓ | FLOPs(T) ↓ | Speed ↑  | sFID ↓ | FID ↓ | Precision ↑ | Recall ↑ |
> |------------------------------|--------------|------------|----------|--------|-------|-------------|----------|
> | DiT-XL/2-G (cfg = 1.50)      | 0.455        | 23.74      | 1.00×    | 4.40   | 2.43  | 0.80        | 0.59     |
> | 50% steps                    | 0.238        | 11.86      | 2.00×    | 4.74   | 3.18  | 0.79        | 0.58     |
> | 40% steps                    | 0.197        | 9.50       | 2.50×    | 5.15   | 3.81  | 0.78        | 0.57     |
> | 34% steps                    | 0.173        | 8.08       | 2.94×    | 5.76   | 4.58  | 0.77        | 0.56     |
> | FORA ($\mathcal{N}=2.5$)     | 0.219        | 10.48      | 2.27×    | 6.59   | 3.83  | 0.79        | 0.55     |
> | FORA ($\mathcal{N}=3$)       | 0.211        | 8.58       | 2.77×    | 6.43   | 3.88  | 0.79        | 0.54     |
> | ToCa ($\mathcal{N}=3, R=0.93$)| 0.227        | 10.23      | 2.32×    | **4.74**   | **3.04**  | **0.80**        | **0.57**     |
> | ToCa ($\mathcal{N}=4, R=0.93$)| 0.209        | 8.73       | 2.72×    | 5.11   | 3.60  | 0.79        | 0.56     |
>
> [1] https://huggingface.co/black-forest-labs/FLUX.1-schnell
>
> [2] Selvaraju P et al. FORA: Fast-Forward Caching in Diffusion Transformer Acceleration. arXiv 2024.

---

> ### Author Response · Authors · 2024-11-26
> **Response to Reviewer R8q5**
>
> Dear Reviewer R8q5,
>
> Thank you very much for your thoughtful and valuable feedback. We have carefully reviewed and addressed the concerns you raised in your list to the best of our ability, making significant adjustments to the manuscript accordingly.
>
> With **the PDF modification deadline just two days away**, we would like to confirm if you have any additional concerns or require further clarification. Please do not hesitate to reach out if there is anything else we can assist with.
>
> Thank you once again for your time and effort throughout this process!

---

> > ### Comment · Reviewer_R8q5 · 2024-11-27
> > **Post-rebuttal Comments**
> >
> > The authors have addressed my major concerns and I will keep my score.

---

> ### Author Response · Authors · 2024-11-27
> **Thanks for supporting acceptance**
>
> Thank you for supporting the acceptance of our paper! Your feedback has been very helpful in refining ToCa and making it a more complete and impactful contribution.
>
> We sincerely appreciate the time and effort you've invested in reviewing our work. If you have any further questions, please feel free to reach out.

---

### Official Review · Reviewer_sYWB · 2024-11-03

**Soundness:** 3
**Presentation:** 3
**Contribution:** 3
**Rating:** 6
**Confidence:** 4

**Summary:**

This paper presents a method called ToCa, which utilizes token-wise feature caching to accelerate the denoising process of DiT. Unlike existing DiT acceleration methods, ToCa offers a fine-grained, token-wise acceleration approach, demonstrating improved performance compared to previous methods.

**Strengths:**

- The paper is well-written and easy to understand.
- The observation that the similarity and propagated error differ for each token in DiT is very insightful.
- Various metrics were proposed to measure the importance of tokens on reusing.
- Experiments were conducted on a variety of datasets, including not only text-to-image but also text-to-video.

**Weaknesses:**

- It's difficult to understand how token-wise caching leads to actual acceleration. Since the attention layer calculates the similarity among all inputs through matrix multiplication, even if one output token is not computed, there won't be a significant difference in the overall computational cost (similar to unstructured pruning). A more detailed explanation of where this acceleration comes from or breakdown is required.
- Evaluation is performed with just single Latency/FID point. An evaluation of the Pareto curve with latency/FID would provide more insightful information.
- There seems to be a color error in Fig 3(b). The yellow arrow indicating low cache ratio is missing.

**Questions:**

Same with my Weakness 1.

---

> ### Author Response · Authors · 2024-11-24
> **Response to Reviewer sYWB Part 1/2**
>
> We sincerely appreciate your valuable feedback and recognition of our work's contribution. We have carefully addressed each of your questions and detailed our responses below.
>
> **W1: It's difficult to understand how token-wise caching leads to actual acceleration. Since the attention layer calculates the similarity among all inputs through matrix multiplication, even if one output token is not computed, there won't be a significant difference in the overall computational cost (similar to unstructured pruning). A more detailed explanation of where this acceleration comes from or breakdown is required.**
>
> *A1:* Thank you for your suggestion. In the updated PDF, we have added a detailed breakdown in the **Appendix A.4 COMPUTATION COMPLEXITY ANALYSIS** section, elaborating on how token-wise caching reduces computational costs and the additional costs associated with token selection.
>
> ToCa leads to real acceleration. Please note that we skip all the computation in the self-attention layers by caching in the default setting of ToCa, which leads to acceleration with no doubt. In MLP layers, we select a set of important tokens for real computation while skipping the other tokens by caching, which directly reduces the number of columns in a matrix, and hence reduces the computation costs of matrix multiplication in MLP layers.
>
> ToCa is not unstructured Pruning. ToCa is a structured acceleration method which don't require any additional support from hardware. The shape of features in diffusion transformers is $R^{N \times D}$, where N denotes the number of tokens while D denotes the dimension of hidden states. In ToCa, we reduce this shape to $R^{M \times D}$ where $M <<N$ denotes the number of tokens not cached. Such a reduction is not applied in any single value of the matrix (unstructured pruning). In contrast, it happens in the column of the matrix, making it become a structured pruning method. ToCa is more likey to be a dual version of channel-wise pruning in traditional parameter pruning methods.
>
> **More detailed explanation in how ToCa achieves acceleration. According to your most helpful comments, we add the following paragraphs to explain how the acceleration of ToCa is achieved in different types of layers.**
>
> **Self-Attention.** Removing a single token in the self-attention calculation affects the dot-product operation between the query and key matrices. If a token is excluded, the query and key matrices each lose one token, leaving $N-1$ tokens in both matrices, where $N$ is the total number of tokens. Consequently, this reduces the computation of vector dot products to $(N-1)^2$, resulting in an $(N-1) \times (N-1)$ self-attention map. This reduction eliminates an entire row and column of computations rather than merely a single element, making the computation reduction structured.
>
> In **self-attention layers**, the major computational costs can be divided into two parts:
>
> - **Linear layers** (from input $x$ to $q, k, v$ and the output projection layer):
>
>    These have a computational complexity of $O(ND^2)$, where $D$ represents the hidden state dimension of each token. Since linear layers operate solely on the hidden states, reducing tokens in a token-wise manner is analogous to reducing the batch size, resulting in a linear reduction in computational cost.
>
> - **Dot-product operations between $q, k, v$:**
>
>    This has a computational complexity of $O(N^2D)$. When the input $x$ reduces by R% of tokens, the number of tokens in $q, k, v$ also decreases by R%. Consequently, the computation of $QK^T$ reduces to $(1-$R%$)^2$ of its original complexity, and the computation for the self-attention map and the dot-product with the value matrix $v$ similarly scales to $(1-$R%$)^2$. Hence, token reduction in self-attention layers is structured and effectively reduces computational costs.
> For **cross-attention layers**, the query $q \in \mathbb{R}^{N \times D}$ is derived from image tokens $x$ through a linear layer, while $k, v \in \mathbb{R}^{N_T \times D}$ are computed from text-conditioning information. Given the number of image tokens $N$ and text tokens $N_T$:
>
> - The linear layers for $q$ and the output projection layer have a complexity of $O(ND^2)$, which linearly reduces with the number of image tokens $N$.
>
> - The linear layers for $k, v$ have a complexity of $O(N_TD^2)$, independent of $N$.
>
> - For the dot-product computations of $q, k, v$ in cross-attention, the complexity is $O(NN_TD)$. Reducing R% of image tokens reduces the corresponding R% $\times N$ rows in the cross-attention map, leading to an R% reduction in computational cost. The same applies to the dot-product between the cross-attention map and $v$. Thus, cross-attention computational costs also decrease linearly and in a structured manner with token reduction.
>
> Due to space constraints, we analyze the rest computational complexity in Response Part 2.

---

> > ### Author Response · Authors · 2024-11-24
> > **Response to Reviewer sYWB Part 2/2**
> >
> > For **MLP layers**, the linear transformations occur along the hidden state dimension and are independent of the token dimension. Reducing the number of tokens results in computational savings similar to reducing the batch size, leading to structured and linear acceleration.
> >
> > In summary, token-wise caching provides structured and effective computational reductions without requiring additional hardware support. Moreover, in practice, we take further steps to manage potential error accumulation in certain layers, such as self-attention. For instance, instead of partially computing tokens, we either compute all tokens or cache all tokens in self-attention layers (see the last paragraph of **Section 3.6** in the main paper: “we propose to cache all tokens in self-attention layers”). Using cached full features directly without computation significantly accelerates these layers.
> >
> > Finally, all our experimental results substantiate that token-wise caching achieves structured computational reduction. The results in **Tables 1, 2, and 3** of the main paper, as well as **Tables 6 and 7** in the appendix, demonstrate that ToCa not only reduces computational costs but also leads to substantial latency improvements, providing strong empirical evidence for its effectiveness.
> >
> > **W2: Evaluation is performed with just single Latency/FID point. An evaluation of the Pareto curve with latency/FID would provide more insightful information.**
> >
> > **A2:** Thank you for your suggestion. Plotting the curves indeed provides a clearer and more comprehensive way to demonstrate the superiority of our ToCa over other methods. Given that latency can vary significantly across different devices and environmental configurations, we have opted to use the more stable FLOPs metric to showcase the acceleration effect. In **Appendix Figure 15**, we present a curve comparison of acceleration methods with 50 steps of DDIM sampling as the baseline. It can be observed that the ToCa curve is below the FORA curve, and both are positioned beneath the curve that directly reduces DDIM sampling steps. This comparison more convincingly demonstrates the effectiveness of ToCa.
> >
> > **W3: There seems to be a color error in Fig 3(b). The yellow arrow indicating low cache ratio is missing.**
> >
> > **A3:** Thank you for your observation. We noticed similar feedback from other reviewers. When viewing the submitted PDF using browsers such as Edge or Chrome, this issue does not appear. However, we identified that certain mobile browsers fail to display the gradient colors in vector graphics correctly. Specifically, the arrow in **Figure 3(b)** is designed to have a yellow lower end transitioning to a blue upper end.

---

> ### Comment · Reviewer_sYWB · 2024-11-26
> **Rebuttal response**
>
> Thank you for the detailed response. The author has clearly addressed most of my concerns. I still believe this paper offers clear advantages over other feature-caching diffusion methods and deserves acceptance at ICLR. I am maintaining my score of 6.

---

> ### Author Response · Authors · 2024-11-26
> **Thanks for your acknowledgment!**
>
> Thank you for your recognition! We are grateful for your acknowledgment that our paper "**deserves acceptance at ICLR**". Your valuable suggestions have greatly helped us further improve the quality of ToCa!

---

### Official Review · Reviewer_3XM7 · 2024-11-03

**Soundness:** 3
**Presentation:** 3
**Contribution:** 3
**Rating:** 6
**Confidence:** 4

**Summary:**

Token-wise Caching (ToCa) is introduced as a fine-grained feature caching strategy designed to accelerate diffusion transformers. It uniquely considers error propagation in caching, utilizing four selection scores to determine the best tokens for caching without incurring extra computational costs. ToCa also allows for variable caching ratios across different layers and integrates multiple techniques to enhance caching efficiency. Extensive experiments on models like PixArt-α, OpenSora, and DiT show that ToCa achieves significant speedups while preserving nearly lossless generation quality.

**Strengths:**

1. The motivation behind the proposed approach is both technically sound and clearly explained, supported by two informative figures illustrating the varying levels of similarity among different tokens and the accumulation of error across these tokens. These visual aids effectively convey the rationale for the proposed method.

2. The methodology for selecting scores and making decisions for each layer is novel, offering valuable insights into the acceleration of transformer models. This innovative approach not only enhances the efficiency of the models but also contributes to a deeper understanding of feature caching strategies within the context of transformers.

**Weaknesses:**

1. There appears to be a lack of a complete algorithmic description within the manuscript. Specifically, a detailed explanation is needed for the token selection process at each layer and each timestep, as well as the procedure for redistributing the cached tokens back into the overall framework. This omission could lead to misunderstandings regarding the efficacy and functionality of the proposed method.

2. The manuscript does not adequately illustrate the computation costs associated with the proposed approach, making it challenging to comprehend the additional expenses involved. A clearer breakdown of these costs would enhance the reader's understanding of the method's efficiency.

3. Including a token diagram for specific layers during both early and late timesteps, across various layer types, may provide additional clarity. Such visual representations could effectively illustrate how tokens are managed and utilized within the different stages of the computation, facilitating a better grasp of the overall caching strategy.

**Questions:**

1. The introductory paragraph of the methods section should be rephrased to clarify the concept of the "naive scheme" for feature caching. This naive scheme is actually no different from existing approaches, such as DeepCache, FORA, and delta-DiT, which should not be "we propose".

2. It should be noted that Figure 3(b) does not depict any low-ratio caching scenarios.

3. There is a discrepancy regarding the application of the scores s1, s2, and s3 across the layers. Specifically, s1 is applicable only to self-attention layers, while s2 pertains solely to cross-attention layers. However, the current definition combines these scores, which raises questions about how this aggregation functions across the different types of layers.

4. Clarification is needed regarding the additional time cost associated with computing the selection scores. Given that the scoring relies on rank computation, there remains a need to calculate and distribute the cached tokens among the uncached tokens. This process is somewhat ambiguous, particularly concerning how the acceleration time is achieved. Presenting the actual extra costs incurred for selecting scores alongside the resulting speedup from the caching process would provide valuable insights.

---

> ### Author Response · Authors · 2024-11-24
> **Response to Reviewer 3XM7 Part 1/4**
>
> Thank you for your recognition of our work and your instructive suggestions. Your feedback has greatly assisted us in making ToCa a more comprehensive piece of work!
>
> **W1: There appears to be a lack of a complete algorithmic description within the manuscript. Specifically, a detailed explanation is needed for the token selection process at each layer and each timestep, as well as the procedure for redistributing the cached tokens back into the overall framework. This omission could lead to misunderstandings regarding the efficacy and functionality of the proposed method.**
>
> **A1:**  Thank you for your suggestion. We agree that the algorithmic analysis of ToCa can be further improved. To address this, we have expanded the algorithm illustration in Figure 3 into a more detailed flow diagram, now presented as **Appendix Figure 12**. Below, we provide the detailed step-by-step process of our ToCa method:
>
> **Step1: Cache Initialization in the first step.**  In the first step of each cache cycle (referred to as the "fresh step", e.g., step t in this example), the denoising model performs full computation for all layers and stores the output features of all layer types in the cache. This step initializes the cache for subsequent steps.
>
> **Step2: Computing Caching Scores for each Token.**  In the subsequent cache steps (e.g., step t-1 in this example), the denoising model performs partial computation. For each token, ToCa computes a caching score based on the formula $\mathcal{S}(x_i)=\mathop{\sum}_{j=1}^4 \lambda_j \cdot s_j(x_i)$, as depicted in Figure 4 and summarized in Section 3.6 of our paper.
>
> **Step3: Selecting Caching Tokens for Acceleration.**   Tokens with low caching scores are selected as cached tokens. These tokens will not be computed by the model in the current step. Instead, their output values are retrieved from the cache, enabling computational acceleration.
>
> **Step4: Computing the Fresh tokens.**  The remaining tokens, referred to as fresh tokens, are sent to the computation layers for processing. The results of these computations are then used to update the cache for their respective positions. At the layer's output, the cached tokens' values are retrieved from the cache and combined with the computed outputs of the fresh tokens. This combined output is then passed to the next layer.
>
> **Step5: Repeating Steps 2–4 Until the End of the Cache Cycle.** Steps 2–4 are repeated iteratively until the cache cycle is complete, at which point a new cache cycle begins.
>
>  Additionally, we provide the **pseudocode algorithm** for ToCa in **Appendix Algorithm 1**.
>
> **W2: The manuscript does not adequately illustrate the computation costs associated with the proposed approach, making it challenging to comprehend the additional expenses involved. A clearer breakdown of these costs would enhance the reader's understanding of the method's efficiency.**
>
> **A2:** Thank you for your suggestion. In the updated PDF, we have added a detailed breakdown in the **Appendix A.4 COMPUTATION COMPLEXITY ANALYSIS** section.
>
> In summary, the computation costs of self-attention score (s1), cross-attention score (s2), caching frequency score & spatial bonus **only account for 0.28%, 0.35%, 0.18% of the whole computation costs, respectively**.
>
> In the following Response Part 2, we introduce the computation complexity for attention and MLP layers, as well as the computation costs for token selection in detail.

---

> ### Author Response · Authors · 2024-11-24
> **Response to Reviewer 3XM7 Part 2/4**
>
> **Attention layer.** In the Attention layer, tokens are first processed through a linear layer to generate queries, keys, and values. Next, the queries and keys are multiplied using a dot product, passed through a softmax function, and then multiplied with the values. Finally, the result is passed through another linear layer to produce the output.
>
> The computational cost of the Self-Attention layer can be expressed as:
>
> $\text{FLOPs}_{SA} \approx N \times D \times 3 \times D \times 2 + N^2 \times D \times 2 + N^2 \times H \times 5 + N^2 \times D \times 2 + N \times D \times D \times 2$
> $= 8 \times N \times D^2 + 4 \times N^2 \times D + 5 \times N^2 \times H \approx O(ND^2) + O(N^2D),$
>
> where $N$ is the number of tokens, $D$ is the hidden state of each token, and $H$ is the number of heads ($H \ll D$).
>
>
> The computational cost of a Cross-Attention layer can be expressed as:
>
> $\text{FLOPs}_{CA} \approx N_1 \times D \times D \times 2 + N_2 \times D \times 2 \times D \times 2 + N_1 \times N_2 \times D \times 2 + N_1 \times N_2 \times H \times 5 + N_1 \times N_2 \times D \times 2 + N_1 \times D \times D \times 2$
> $= 4 \times (N_1 + N_2) \times D^2 + 4 \times N_1 \times N_2 \times D + 5 \times N_1 \times N_2 \times H$
> $\approx O((N_1 + N_2)D^2) + O(N_1 N_2D) = O((N + N_2)D^2) + O(N N_2D),$
>
> where $N_1 = N$, $N_2$ are the number of image and text tokens, $D$ is the hidden state of each token, and $H$ is the number of heads ($H \ll D$). In the previous computations, the softmax operation is approximated as involving 5 floating-point calculations per element.
>
> **MLP layer.** The computational cost of the MLP layer can be written as:
>
> $\text{FLOPs}_{MLP} \approx N \times D_1 \times D_2 \times 2 + N \times D_2 \times 6 + N \times D_1 \times D_2 \times 2$
> $= 4 \times N \times D_1 \times D_2 + 6 \times D_2 \times N$
> $= 16 \times N \times D^2 + 24 \times N \times D \approx O(ND^2),$
>
> where $N$ is the number of tokens, $D_1 = D$, and $D_2 = 4D_1$ are the hidden and middle-hidden states in the MLP, respectively. The activation function for the MLP is approximated to involve 6 floating-point operations per element.
>
> Besides, there are computational costs by Token Selection process, as analyzed below:
>
> **Self-Attention Score $s_1$.** As mentioned in section 3.5, the Self-Attention score $s_1$ is computed as $s_1(x_i) = \sum_{i=1}^{N} \alpha_{ij}$, where $x_i$ is the $i_{th}$ token, and $\alpha_{ij}$ is the element in the self-attention map. Therefore, the computational complexity of the self-attention score is only $N \approx O(N)$. In a practical case achieving about $2.3\times$ acceleration, the computation cost of the Self-Attention score in TOCA accounts for approximately $0.28$% of the main components.
>
> **Cross-Attention Score $s_2$.** As mentioned in section 3.5 the Cross-Attention score $s_2$ is computed as $s_2(x_i) = -\sum_{j=1}^{N} c_{ij} \log (c_{ij})$,  where the $c_{ij}$ is the element in the cross-attention map. Therefore, the computational complexity of the cross-attention score is only $2N \approx O(N)$. In a practical case achieving about $2.3\times$ acceleration, the computation cost of the Cross-Attention score in TOCA accounts for approximately $0.35$% of the main components.
>
> **Cache Frequency Score $s_3$ and Uniform Spatial Distribution Score $s_4$.** The Cache Frequency score $s_3$ is updated at each step, so its update cost per timestep is $N$. When the Cache Frequency score is called for summation in practical applications, the computation cost is $2N$. Thus, the total cost for one layer is $3N \approx O(N)$. The Uniform Spatial Distribution score $s_4$ is computed by sorting within each grid of size $G \times G$ and weighting the top-scoring tokens. The computation cost is given by $\frac{N}{G^2} \times G^2 \log(G^2)  + 2 \times \frac{N}{G^2}$, where $G$ is the grid size, which is usually small. Therefore the computational complexity of $s_4$ is $O(N)$. In a practical case achieving about $2.3\times$ acceleration, the computation cost of the Cache Frequency score $s_3$ accounts for approximately $0.044\%$  and the Uniform Spatial Distribution score $s_4$ accounts for $0.15$% of the main computation components. In addition, the computational cost for sorting $N$ tokens is $O(N\log N)$, which accounts for approximately $0.18$% of the main computational cost.
>
> Due to space constraints, we summary the additional computations in Response Part 3.

---

> ### Author Response · Authors · 2024-11-24
> **Response to Reviewer 3XM7 Part 3/4**
>
> In summary, although ToCa introduces additional computations, its computational complexity of $O(N)$ is negligible compared to the main computational modules with complexities of $O(N^2 D)$ or $O(N D^2)$. In practical tests, the time taken for token selection is minimal, typically less than $1$% of the main computational cost. At cache steps, taking a caching ratio of $\mathcal{R}=90$% as an example, the computational cost of terms with a complexity of $O(N D^2)$ is reduced to $10$% of the original, while the computational cost of terms with a complexity of $O(N^2 D)$ is reduced to $1$%. As mentioned earlier, in practice, it is more efficient to shift all computations at cache steps to the MLP. Therefore, all terms with a complexity of $O(N^2 D)$ at cache steps are ignored.
>
> **W3: Including a token diagram for specific layers during both early and late timesteps, across various layer types, may provide additional clarity. Such visual representations could effectively illustrate how tokens are managed and utilized within the different stages of the computation, facilitating a better grasp of the overall caching strategy.**
>
> **A3:** Thank you for your insightful suggestion. Adding a token diagram is indeed a more intuitive way to convey the caching strategy compared to text and equations alone. In the updated **Appendix Figure 14**, we provide a detailed illustration of the computation allocation across different layer types, depths, and timesteps using DiT as an example.
>
> As described in our paper, due to the cost-effectiveness of computation allocation, no additional computation is assigned to the self-attention layers during cache steps. Thus, as shown in the diagram, computation is only allocated during the fresh steps, while no computation is performed during cache steps. In contrast, more computation is allocated to the MLP layers, where the allocation gradually increases as timesteps decrease (from large to small) and decreases as the layer depth increases. This trend is clearly reflected in the diagram.
>
> For text-conditional models, which include cross-attention layers, the computation allocation for cross-attention layers lies between that of MLP layers and self-attention layers.
>
> We believe this visual presentation makes it easier for readers to understand how tokens are managed and computation is distributed in our ToCa method. Thank you again for your valuable suggestion!
>
> **Q1: The introductory paragraph of the methods section should be rephrased to clarify the concept of the "naive scheme" for feature caching. This naive scheme is actually no different from existing approaches, such as DeepCache, FORA, and delta-DiT, which should not be "we propose".**
>
> **A4:**  Thank you for pointing this out. We agree that the original phrasing in the draft was imprecise. In the updated PDF version, we have revised this section to replace the previous statement:*"Following previous caching methods for U-Net-like diffusion denoisers, we introduce the naive scheme for feature caching."* with:*"We follow the naive scheme for feature caching adopted by most previous caching methods such as DeepCache [1], FORA [2], and $\Delta$-DiT [3] for diffusion denoisers."* We have also added the appropriate citations to ensure proper acknowledgment of prior works.
>
> [1] Ma et al. Deepcache: Accelerating diffusion models for free. CVPR 2024.
>
> [2] Selvaraju P et al. FORA: Fast-Forward Caching in Diffusion Transformer Acceleration. arXiv 2024.
>
> [3] Chen et al. $\Delta$-DiT: A Training-Free Acceleration Method Tailored for Diffusion Transformers. arXiv 2024.
>
> **Q2: It should be noted that Figure 3(b) does not depict any low-ratio caching scenarios.**
>
> **A5:** Thank you for your observation. We noticed similar feedback from other reviewers. When viewing the submitted PDF using browsers such as Edge or Chrome, this issue does not appear. However, we identified that certain mobile browsers fail to display the gradient colors in vector graphics correctly. Specifically, the arrow in **Figure 3(b)** is designed to have a yellow lower end transitioning to a blue upper end.
>
> To address this issue, we have updated the figure in the revised version by adjusting the gradient settings of the vector graphic and embedding the problematic arrow as a standalone PNG image. The updated version should now correctly display the arrow with a yellow lower end transitioning to light blue at the upper end across all platforms.

---

> ### Author Response · Authors · 2024-11-24
> **Response to Reviewer 3XM7 Part 4/4**
>
> **Q3: There is a discrepancy regarding the application of the scores s1, s2, and s3 across the layers. Specifically, s1 is applicable only to self-attention layers, while s2 pertains solely to cross-attention layers. However, the current definition combines these scores, which raises questions about how this aggregation functions across the different types of layers.**
>
> **A6:** Thank you for raising this important point. You are correct that in practice, $s_1$ is calculated solely through self-attention layers, $s_2$ is only calculated through cross-attention layers, and $s_3$ is updated across all layers. This design raises the question of how these scores are utilized in layers such as MLP layers, where $s_1$, $s_2$, and $s_3$ are not directly computed.
>
> To clarify, in our implementation, the calculation materials for $s_1$ and $s_2$, which are based on the self-attention or cross-attention features, are cached along with the corresponding attention maps during the caching process. Since attention maps occupy a space comparable to a feature map, the additional storage required for precalculating $s_1$ and $s_2$ is minimal. Both $s_1$ and $s_2$ are precomputed during the self-attention and cross-attention computations, respectively, and stored as vectors of length-$N$ , where $N$ is the total number of tokens.
>
> For $s_3$, it is stored as alength-$N$ counter, where each element tracks the number of consecutive cache steps a token has been cached. This additional memory overhead is negligible compared to the feature maps themselves.
>
> During cache steps, when token scores are required, the precomputed scores $s_1$, $s_2$, and $s_3$ are directly retrieved from the cache and aggregated using the formula $\sum_{j=1}^3 \lambda_j \cdot s_j(x_i)$.
>
> The spatial distribution adjustment $s_4$ is then applied to compute the final importance score for each token. Based on these scores, tokens are divided into those requiring computation (i.e., fresh tokens) and those that can remain cached (i.e., cached tokens).
>
> We have added this clarification to the revised version of the manuscript to ensure transparency regarding how token scores are aggregated and utilized across different layer types.
>
>
> **Q4: Clarification is needed regarding the additional time cost associated with computing the selection scores. Given that the scoring relies on rank computation, there remains a need to calculate and distribute the cached tokens among the uncached tokens. This process is somewhat ambiguous, particularly concerning how the acceleration time is achieved. Presenting the actual extra costs incurred for selecting scores alongside the resulting speedup from the caching process would provide valuable insights.**
>
> **A7:** We have made a detailed computation breakdown in **A2**.

---

> ### Author Response · Authors · 2024-11-26
> **Response to Reviewer 3XM7**
>
> Dear Reviewer 3XM7,
>
> Thank you once again for your constructive and detailed feedback. We have carefully revised the manuscript following your recommendations. With **the PDF modification deadline approaching in less than two days**, we are yet to hear further feedback on our rebuttal. Please feel free to reach out if you require any clarification or additional information—we are ready to provide it promptly!

---

> ### Author Response · Authors · 2024-11-28
> **Response to Reviewer 3XM7**
>
> Dear Reviewer 3XM7,
>
> Thank you once again for your thoughtful and detailed suggestions. In the latest version of the PDF, we have carefully incorporated the adjustments based on your recommendations. As **the deadline for the discussion phase approaches**, we have addressed most of the concerns raised by the other reviewers, but we have not yet received further feedback from you. We would like to confirm whether our responses have fully addressed your concerns, or if there are any remaining areas that may require further refinement.
>
> We truly appreciate the time and effort you have dedicated to reviewing our work, and we look forward to hearing from you!

---

> ### Author Response · Authors · 2024-11-30
> **Response to Reviewer 3XM7**
>
> Dear Reviewer 3XM7,
>
> We would like to sincerely thank you once again for your thoughtful and insightful suggestions. In the latest version of the PDF, we have made careful adjustments based on your valuable feedback. As **the discussion phase deadline approaches**, we have addressed almost all of the concerns raised by the other reviewers, but we have not yet had the pleasure of receiving your response.
>
> We would be truly grateful if you could let us know whether our revisions have sufficiently addressed your concerns, or if there are any remaining areas that may still need further refinement.
>
> We greatly appreciate the time and effort you have dedicated to reviewing our work, and we are hopeful to hear from you soon.

---

> > ### Comment · Reviewer_3XM7 · 2024-12-03
> > **Reviewer Response**
> >
> > I appreciate your thorough response, which included a comprehensive explanation of the methodology, a detailed computational cost analysis, and additional insights. Your responses have largely alleviated my concerns, and as a result, I have adjusted my assessment score accordingly.

---

> > > ### Author Response · Authors · 2024-12-03
> > > **Thanks for your support!**
> > >
> > > Thank you for your thoughtful feedback and for recognizing the **comprehensive explanation of the methodology**, the **detailed computational cost analysis**, and the **additional insights** we provided. We are glad that our responses have addressed your concerns and contributed to your reassessment.
> > >
> > > We truly appreciate your constructive suggestions, which have been very helpful in refining and improving our work. Thank you again for your time and support!

---

### Official Review · Reviewer_Vwce · 2024-11-04

**Soundness:** 3
**Presentation:** 3
**Contribution:** 2
**Rating:** 6
**Confidence:** 5

**Summary:**

The paper introduces ToCa (Token-wise feature Caching), a training-free feature caching method tailored to accelerate diffusion transformers. ToCa allows for adaptive token selection, which aims to cache tokens with minimal caching error based on various criterions. Experiments have been conducted on PixArt-α, OpenSora, and DiT models to show the speedup and quality trade-off of ToCa.

**Strengths:**

- The visualizations of temporal redundancy and error propagation illustrate the motivation of ToCa in an intuitive way, making the design choices relatively well motivated.
- The four token selection scoring functions and layer-specific cache ratios are natural design choices that enable the caching strategy to be more fine-grained
- ToCa achieves more than 2x acceleration ratios on both text-to-image and text-to-video tasks, while having better quality than baselines.
- ToCa’s training-free nature makes it more practical for real-world application.

**Weaknesses:**

- While ToCa achieves reasonable benchmark results, some artifacts remain in the generated images compared to the originals. For instance, in Figure 6, the moon is missing from the "wolf howling at the full moon" prompt, and the background forests appear blurred in the "tranquil beach" prompt. It is necessary to demonstrate how ToCa performs with high-resolution images (1024x1024) generated by more advanced models like FLUX.1-dev [1].
- Another important direction in accelerating diffusion models involves reducing the required sampling steps. However, this might compromise ToCa's effectiveness since fewer steps may result in greater feature variations between steps, potentially making them less suitable for caching. Please include experimental results using models with reduced sampling steps, such as FLUX.1-schnell [2] with 4 steps.
- For advanced model evaluation, FID and CLIP scores may not adequately reflect human preferences. Please provide quantitative results using more recent metrics, such as image reward [3].
- The method involves numerous hyperparameters, including four token selection scoring metrics ($\lambda_1$, $\lambda_2$, $\lambda_3$, $\lambda_4$) and caching ratios for different layers and timesteps ($\lambda_l$, $\lambda_{type}$, $\lambda_t$). While A.3 provides a sensitivity analysis, it remains unclear how these parameters should be determined for new models or tasks. The authors should elaborate on hyperparameter selection and impacts, particularly addressing whether an automatic selection method (e.g., a calibration procedure) exists. The complexity of hyperparameters could limit the method's generalizability and practical application.
- The "Acceleration of Diffusion Models" section lacks references to key fundamental works. Specifically, step distillation [4, 5] and consistency models [6] should be cited under "reducing the number of sampling timesteps," and Q-Diffusion [7] should be cited under "weight quantization."
- Notation inconsistencies: s3 is generally defined as cache frequency but appears as spatial distribution in Table 4, which uses s4 for cache frequency.

[1] https://huggingface.co/black-forest-labs/FLUX.1-dev
[2] https://huggingface.co/black-forest-labs/FLUX.1-schnell
[3] Xu et al. ImageReward: Learning and Evaluating Human Preferences for Text-to-image Generation. NeurIPS 2023.
[4] Salimans et al. Progressive Distillation for Fast Sampling of Diffusion Models. ICLR 2022.
[5] Meng et al. On Distillation of Guided Diffusion Models. CVPR 2023.
[6] Song et al. Consistency Models. ICML 2023.
[7] Li et al. Q-Diffusion: Quantizing Diffusion Models. ICCV 2023.

**Questions:**

- In table 2, ToCa does not outperform PAB with 1.24x speedup (78.34 vs 78.51). Although ToCA achieves a more significant speed-up, it is crucial to preserve the generation quality. I am wondering if ToCa can have better generation quality using a more conservative setting that has similar speed-up to PAB?

---

> ### Author Response · Authors · 2024-11-24
> **Response to Reviewer Vwce Part 1/4**
>
> Thank you very much for your positive feedback and constructive suggestions. Your professional advice has been invaluable in further improving our work!
>
> **W1: While ToCa achieves reasonable benchmark results, some artifacts remain in the generated images compared to the originals. For instance, in Figure 6, the moon is missing from the "wolf howling at the full moon" prompt, and the background forests appear blurred in the "tranquil beach" prompt. It is necessary to demonstrate how ToCa performs with high-resolution images (1024x1024) generated by more advanced models like FLUX.1-dev.**
>
> **W2: Another important direction in accelerating diffusion models involves reducing the required sampling steps. However, this might compromise ToCa's effectiveness since fewer steps may result in greater feature variations between steps, potentially making them less suitable for caching. Please include experimental results using models with reduced sampling steps, such as FLUX.1-schnell with 4 steps.**
>
> **W3: For advanced model evaluation, FID and CLIP scores may not adequately reflect human preferences. Please provide quantitative results using more recent metrics, such as image reward.**
>
> **A1:** As you pointed out, applying our ToCa to high-resolution generation with more advanced models like FLUX evaluated by more recent metrics like image reward can further highlight the advantages of ToCa. Below, we address **W1-W3** by two additional experimental results:
>
> **1. TOCA outperforms previous methods on FLUX.**
>
> In the latest version of our paper, we have added experimental results in Appendix Table 7 (Table 1 below) to compare ToCa with the baseline method FORA [1] on the FLUX.1-schnell model [2]. For evaluation, we adopted *Image Reward* [3], which better reflects human preferences. The results demonstrate that ToCa achieves **1.5× lossless acceleration**. In contrast, FORA achieves only 1.2× acceleration, with the Image Reward dropping by 0.004 from the baseline value of 1.133. For ToCa at 1.5× speedup, the *Image Reward* remains almost unchanged from the baseline value of 1.133. Additionally, we present corresponding visual examples in **Appendix Figure 13**, where the samples accelerated by ToCa at 1.5× are nearly indistinguishable from the original unaccelerated samples.
>
> **Table1: Quantitative comparison in text-to-image generation for FLUX on Image Reward. FORA$^{1}$, FORA$^{2}$, and FORA$^{3}$ correspond to skipping the 2nd, 3rd, and 4th steps, respectively, during the 4-step generation process.**
>
> | Method                          | Latency(s) ↓ | FLOPs(T) ↓ | Speed ↑ | Image Reward ↑ |
> |---------------------------------|--------------|------------|---------|----------------|
> | FLUX.1-schnell                  | 2.882        | 277.88     | 1.00×   | 1.133          |
> | FORA¹                           | 2.365        | 225.60     | 1.23×   | 1.129          |
> | FORA²                           | 2.365        | 225.60     | 1.23×   | 1.124          |
> | FORA³                           | 2.365        | 225.60     | 1.23×   | 1.123          |
> | ToCa ($\mathcal{N}=2,R=0.90$)   | **1.890**        | **181.30**    | **1.53×**   | **1.134**          |
>
> Due to space constraints, we include the second experimental result in Response Part 2.

---

> ### Author Response · Authors · 2024-11-24
> **Response to Reviewer Vwce Part 2/4**
>
> **2. ToCa demonstrates strong performance with low sampling steps in both FLUX and DiT models.**
>
> As shown in the Table 1 above, we have already demonstrated that ToCa performs well even with 4-step inference in FLUX. Additionally, in Appendix Table 6 (Table 2 below), we provide new experimental results on Diffusion Transformer (DiT) with 50 DDIM sampling steps, comparing it to the previous DiT experiments with 250 sampling steps. The results in Table 2 show that, under 50 sampling steps, ToCa with 2.7× acceleration increases the FID-50k from 2.58 (at 250 sampling steps) to 3.60. Similarly, ToCa with 2.3× acceleration achieves an FID-50k of 3.04. This indicates that reducing the number of sampling steps does lead to a drop in generation quality. However, even with the reduced 50 sampling steps, ToCa still significantly outperforms both FORA and the approach of simply reducing sampling steps in terms of generation quality.
>
> These new findings further demonstrate the superiority of our ToCa in finer-grained computation allocation strategy.
>
> **Table2: Quantitative comparison on class-to-image generation on ImageNet with 50 steps DDIM sampler as baseline on DiT-XL/2.**
>
> | Method                       | Latency(s) ↓ | FLOPs(T) ↓ | Speed ↑  | sFID ↓ | FID ↓ | Precision ↑ | Recall ↑ |
> |------------------------------|--------------|------------|----------|--------|-------|-------------|----------|
> | DiT-XL/2-G (cfg = 1.50)      | 0.455        | 23.74      | 1.00×    | 4.40   | 2.43  | 0.80        | 0.59     |
> | 50% steps                    | 0.238        | 11.86      | 2.00×    | 4.74   | 3.18  | 0.79        | 0.58     |
> | 40% steps                    | 0.197        | 9.50       | 2.50×    | 5.15   | 3.81  | 0.78        | 0.57     |
> | 34% steps                    | 0.173        | 8.08       | 2.94×    | 5.76   | 4.58  | 0.77        | 0.56     |
> | FORA ($\mathcal{N}=2.5$)     | 0.219        | 10.48      | 2.27×    | 6.59   | 3.83  | 0.79        | 0.55     |
> | FORA ($\mathcal{N}=3$)       | 0.211        | 8.58       | 2.77×    | 6.43   | 3.88  | 0.79        | 0.54     |
> | ToCa ($\mathcal{N}=3, R=0.93$)| 0.227        | 10.23      | 2.32×    | **4.74**   | **3.04**  | **0.80**        | **0.57**     |
> | ToCa ($\mathcal{N}=4, R=0.93$)| 0.209        | 8.73       | 2.72×    | 5.11   | 3.60  | 0.79        | 0.56     |
>
> [1] Selvaraju P et al. FORA: Fast-Forward Caching in Diffusion Transformer Acceleration. arXiv 2024.
>
> [2] https://huggingface.co/black-forest-labs/FLUX.1-schnell
>
> [3] Xu et al. ImageReward: Learning and Evaluating Human Preferences for Text-to-image Generation. NeurIPS 2023.

---

> ### Author Response · Authors · 2024-11-24
> **Response to Reviewer Vwce Part 3/4**
>
> **W4: The method involves numerous hyperparameters, including four token selection scoring metrics (λ1, λ2, λ3, λ4) and caching ratios for different layers and timesteps (λl, λtype, λt). While A.3 provides a sensitivity analysis, it remains unclear how these parameters should be determined for new models or tasks. The authors should elaborate on hyperparameter selection and impacts, particularly addressing whether an automatic selection method (e.g., a calibration procedure) exists. The complexity of hyperparameters could limit the method's generalizability and practical application.**
>
> **A2:** Thank you for your valuable suggestion. We would like to confirm that **TOCA is not highly sensitive to hyperparameters. Our experiment results show that the same hyper-parameter works well on all the models of the same tasks (text-to-image, text-to-video), consistently outperforming other previous methods**, as elaborated below.
>
> As discussed in **Section 3 Methodology** of our paper, the hyperparameters $\lambda_1$ and $\lambda_2$ represent the weights for self-attention and cross-attention scores, respectively. For example, in the class-conditional generation model DiT, which lacks a cross-attention module, $\lambda_2$ is naturally set to 0, while $\lambda_1$ is fixed at 1. In models like PixArt and OpenSora that include both self-attention and cross-attention, the choice of $\lambda_1$ and $\lambda_2$ reflects the balance between different model aspects. As shown in Table 5 of our paper, using self-attention scores only ($\lambda_1 = 1$, $\lambda_2=0$) results in better FID scores but lower CLIP scores, suggesting that the model better captures inter-token relationships at the cost of controllability. Conversely, focusing only on cross-attention scores ($\lambda_1 = 0$, $\lambda_2=1$) slightly worsens FID but improves CLIP scores, indicating better alignment with text conditions. Hence, the choice of $\lambda_1$ and $\lambda_2$ should align with the desired generation style, and we recommend using a larger $\lambda_2$ value in models with cross-attention models for improved text-image alignment, as the overall generation quality remains largely unaffected (see **Table 5** in our paper).
>
> The hyperparameters $\lambda_3$ and $\lambda_4$ control the weights for the cache frequency score and uniform spatial distribution score, respectively, to prevent excessive caching from the temporal and spatial perspectives. As shown in the sensitivity analysis in **Appendix Figure 11(d)**, $\lambda_3$ only affects generation quality when it is very small. As $\lambda_3$ increases, FID improves until a point (0.25 in our experiments) where it stabilizes and becomes insensitive, indicating that the model has effectively adjusted token selection based on cache frequency. At this point, cached tokens are quickly updated, avoiding local error accumulation and preserving global information. Hence, we recommend setting $\lambda_3$ to 0.25 in practice. For $\lambda_4$, which corresponds to the uniform spatial distribution score, **Appendix Figure 11(e)** shows that it has little impact on FID and is therefore an insensitive hyperparameter that can even be disabled in practical applications.
>
> Due to space constraints, we include the rest of analysis in Response Part 4.

---

> ### Author Response · Authors · 2024-11-24
> **Response to Reviewer Vwce Part 4/4**
>
> Hyperparameters related to caching ratios, such as $\lambda_l$, $\lambda_t$, and $\lambda_{type}$, determine the allocation of computation across layers, timesteps, and layer types, respectively. As shown in **Appendix Figure 11**, $\lambda_l$ significantly affects generation quality but has a flat optimal region, meaning that minor deviations from the optimal value have minimal impact on performance. $\lambda_{type}$, which controls allocation across MLP and attention layers, strongly influences generation quality and highlights the importance of prioritizing computation in MLP layers over attention layers. This behavior is consistent across models and tasks, meaning $\lambda_{type}$ generally does not require task-specific tuning. Similarly, $\lambda_t$ is an insensitive parameter.
>
> For the same task and model, the hyperparameters can be fully reused across different sampling step settings, ensuring near-optimal performance. For different tasks or models, minor adjustments to certain hyperparameters, such as $\lambda_l$, may be required to achieve optimal performance, but the impact of such tuning is minimal. Overall, in DiT-based models, the proposed hyperparameters can largely be reused with the same configuration, consistently outperforming other methods.
>
> During our experiments, we observed that all hyperparameters independently influence generation quality. Consequently, applying our ToCa on a new model does not require exhaustive grid search across all hyperparameters. Instead, hyperparameters can be individually adjusted within recommended ranges, significantly reducing tuning complexity. Based on your suggestion, we plan to develop an automatic hyperparameter calibration mechanism leveraging this independence. Each hyperparameter can be treated as a one-dimensional optimization problem within its recommended range, avoiding the need for high-dimensional optimization. Nevertheless, in most cases, using the default hyperparameter configuration is sufficient to achieve excellent generation performance.
>
> **W5: The "Acceleration of Diffusion Models" section lacks references to key fundamental works. Specifically, step distillation and consistency models hould be cited under "reducing the number of sampling timesteps," and Q-Diffusion should be cited under "weight quantization."**
>
> **A3:** Thank you for pointing this out, and we sincerely apologize for overlooking these key works in our original manuscript. In the updated version, we have added citations for step distillation and consistency models under the "reducing the number of sampling timesteps" section, as well as Q-Diffusion under the "weight quantization" section. We appreciate your suggestion, which has helped make our work more complete.
>
> **W6: Notation inconsistencies: s3 is generally defined as cache frequency but appears as spatial distribution in Table 4, which uses s4 for cache frequency.**
>
> **A4:**  We apologize for the oversight, and thank you for pointing this out! In the updated **Table 4**, we have corrected this inconsistency to ensure the notations are accurate and consistent.
>
> **Q1: In table 2, ToCa does not outperform PAB with 1.24x speedup (78.34 vs 78.51). Although ToCA achieves a more significant speed-up, it is crucial to preserve the generation quality. I am wondering if ToCa can have better generation quality using a more conservative setting that has similar speed-up to PAB?**
>
> **A5:**  Thank you for your suggestion. In the latest version of the paper, we have added results comparing ToCa with PAB under a more conservative configuration. The experimental results show that ToCa indeed achieves better performance in this setting. As shown in the updated **Table 2**, with R=80%, ToCa achieves a VBench score of 78.59, surpassing PAB's 78.51, while also achieving a significantly higher speedup of 2.28× compared to PAB's 1.24×. This demonstrates that our method is more effective than PAB, even when preserving generation quality.
>
> In addition, we measured the FLOPs-FID Pareto curve in Appendix Figure 15, using 50 DDIM sampling steps as the baseline, to evaluate the trade-off between acceleration and generation quality achieved by ToCa.

---

> ### Author Response · Authors · 2024-11-26
> **Response to Reviewer Vwce**
>
> Dear Reviewer Vwce,
>
> Thank you once again for your professional and high-quality feedback. We have recently made adjustments to the manuscript based on your insightful suggestions. With **less than two days remaining until the PDF modification deadline**, we have yet to receive further responses to our rebuttal.
>
> If additional information or clarification is needed, please do not hesitate to reach out. We hope to do our best to provide all the necessary details!

---

> ### Author Response · Authors · 2024-11-27
> **Additional experimental results for “Response to Reviewer Vwce Part 1/4”**
>
> Dear Reviewer Vwce,
>
> We have **further supplemented our previous work with experimental results on the advanced model FLUX.1-dev**, which uses 50 sampling steps and a resolution of $1024 \times 1024$. The updated experiments further demonstrate that ToCa can achieve lossless acceleration on high-resolution models like FLUX, outperforming other methods. Please refer to the updated Table 1. Additionally, Table 7 in the PDF has also been updated accordingly.
>
>
> In the latest version of our paper, we have added experimental results in Appendix Table 7 (Table 1 below) comparing ToCa with the baseline method FORA [1] on both the FLUX.1-dev and FLUX.1-schnell models [2]. For evaluation, we adopted *Image Reward* [3], which better reflects human preferences. The results demonstrate that ToCa achieves **1.5× lossless acceleration on both FLUX.1-dev and FLUX.1-schnell**. In contrast, FORA, under the same 1.5× speedup on the FLUX.1-dev model, shows a 0.006 drop in Image Reward from the baseline value of 1.202, while it achieves only 1.2× acceleration on FLUX.1-schnell, with the Image Reward dropping by 0.004 from the baseline value of 1.133. For ToCa at 1.5× speedup, the *Image Reward* remains almost unchanged from the baseline values: 1.202 on FLUX.1-dev and 1.133 on FLUX.1-schnell. Additionally, we present corresponding visual examples for FLUX.1-schnell in **Appendix Figure 13**, where the samples accelerated by ToCa at 1.5× are nearly indistinguishable from the original unaccelerated samples.
>
> **Table1: Quantitative comparison in text-to-image generation for FLUX on Image Reward. FORA$^{1}$, FORA$^{2}$, and FORA$^{3}$ correspond to skipping the 2nd, 3rd, and 4th steps, respectively, during the 4-step generation process.**
>
> | Method                          | Latency(s) ↓ | FLOPs(T) ↓ | Speed ↑ | Image Reward ↑ |
> |----------------------------------|--------------|------------|---------|----------------|
> | FLUX.1-dev     | 33.85        | 3719.50    | 1.00×   | 1.202          |
> | 68% steps                        | 23.02        | 2529.26    | 1.47×   | 1.200          |
> | FORA | 20.82        | 2483.32    | 1.51×   | 1.196          |
> | ToCa ($\mathcal{N}=2, R=0.90$) | **19.88** | **2458.06** | **1.51×** | **1.202**  |
> ||||||
> | FLUX.1-schnell   | 2.882        | 277.88     | 1.00×   | 1.133          |
> | 75% steps                        | 2.162        | 208.41     | 1.33×   | 1.139          |
> | FORA$^1$  | 2.365        | 225.60     | 1.23×   | 1.129          |
> | FORA$^2$| 2.365        | 225.60     | 1.23×   | 1.124          |
> | FORA$^3$  | 2.365        | 225.60     | 1.23×   | 1.123          |
> | ToCa ($\mathcal{N}=2, R=0.90$)   | **1.890**    | **181.30** | **1.53×** | **1.134**      |
>
> With **less than one day remaining before the PDF modification deadline**, we would like to confirm if you have any additional concerns or require further clarifications based on the recent updates. Please do not hesitate to reach out if there is anything else we can assist with before the deadline. Thank you again for your time and valuable input throughout this process!
>
>
> [1] Selvaraju P et al. FORA: Fast-Forward Caching in Diffusion Transformer Acceleration. arXiv 2024.
>
> [2] https://huggingface.co/black-forest-labs/FLUX.1-schnell
>
> [3] Xu et al. ImageReward: Learning and Evaluating Human Preferences for Text-to-image Generation. NeurIPS 2023.thods.

---

> > ### Comment · Reviewer_Vwce · 2024-11-28
> > **Response**
> >
> > Thank you for your response. Most of my concerns have been addressed, and I find the Flux results and Appendix Figure 13 impressive. That said, the figures are not entirely indistinguishable. For example, the text in the sample “A Vietnam map” became less accurate after applying ToCa. However, I acknowledge that this is a particularly challenging sample, as text generation is inherently difficult. This issue does not distract from the overall merit of the paper. I suggest the authors consider addressing such fine-grained details in future work or note them as a limitation. I have raised my scores accordingly.

---

> > > ### Author Response · Authors · 2024-11-28
> > > **Thanks for your suggrestions and support of our work!**
> > >
> > > Thank you for your thoughtful and detailed suggestions, as well as for your support of our work! The performance of ToCa on extremely high-challenge samples warrants further investigation, and we will continue to explore solutions in our future work. We greatly appreciate your recognition of our efforts, and we hope that ToCa will further contribute to the advancement of Diffusion Transformers Acceleration!

---

### Author Response · Authors · 2024-12-03
**Thanks to All Reviewers for Their Thoughtful Feedback and Support**

We sincerely thank all the reviewers for their meticulous and insightful suggestions, as well as for the generous time and effort they devoted during the rebuttal process. Through our collective efforts, the ToCa work has been further improved, including but not limited to additional experiments to enrich the content and more detailed explanations of computational requirements and processes.

**Reviewer Vwce** raised the rating **from 5 to 6** following the rebuttal process. The reviewer primarily suggested conducting further high-resolution image generation experiments on a more advanced model, FLUX, and evaluating the results using the ImageReward metric, which better reflects human preferences, to further explore the potential of ToCa. During the rebuttal period, we addressed these concerns with additional experiments. As the reviewer mentioned, *"I find the Flux results and Appendix Figure 13 impressive,"* ToCa demonstrated **strong performance even in high-resolution settings with advanced models like FLUX**, further supporting the broad applicability of the proposed acceleration method. Reviewer Vwce subsequently increased the score to **support the acceptance** of ToCa.

**Reviewer 3XM7** increased the rating **from 5 to 6** following the rebuttal process. The reviewer primarily focused on the detailed explanation of the algorithm and the analysis of the associated computational costs. During the rebuttal period, we provided further clarifications and supplementary analyses, which received high recognition and support. As the reviewer stated, *"I appreciate your thorough response, which included **a comprehensive explanation of the methodology**, **a detailed computational cost analysis**, and **additional insights**. Your responses have largely alleviated my concerns, and as a result, I have adjusted my assessment score accordingly."*

**Reviewer sYWB** maintained the **rating of 6** and believes our work **"deserves acceptance at ICLR"**. The reviewer’s main concerns were related to further clarification of the acceleration principles behind ToCa and the use of a Pareto curve comparing computational cost and model accuracy to demonstrate ToCa's superiority in various scenarios. During the rebuttal period, we addressed these concerns, and the reviewer also highly praised our work, stating, *"I still believe this paper offers **clear advantages over other feature-caching diffusion methods** and **deserves acceptance at ICLR**."*

**Reviewer R8q5** decided to maintain the **rating of 6** following the rebuttal. The reviewer’s primary concerns focused on a more explicit explanation of the caching method and more comprehensive and convincing experimental evidence. During the rebuttal period, we addressed these concerns, and the reviewer chose to uphold their support for ToCa, stating, *"The authors **have addressed my major concerns**."*

We are thrilled that **all four reviewers have unanimously expressed that this paper is worthy of acceptance**. Once again, we sincerely thank you for your support of our work and the generous time and effort you have devoted to this process!

---

### Meta-Review · Area_Chair_ggS6 · 2024-12-17

**Metareview:**

Summary:
Proposes a token-wise feature caching (ToCa) mechanism for inference-time acceleration of diffusion transformers. ToCa presents an adaptive token selection algorithm, which dynamically selects tokens for caching based on various error criteria. The method is well motivated and supported by strong results, and detailed analysis, showing speedups and preservation of image quality.
Strength:
The problem formulation. The temporal redundancy and error propagation illustrations clearly motivate the design choices. The proposed method for token selection score has substantial novelty, and provides insights into the distribution of features in denoising transformers. Strong results (2x speedup) and a highly desirable achievement for inference-time acceleration, which is another avenue that’s less explored than distillation-based approaches.
Weakness:
The paper at times can be a bit hard to follow with the main contributions distributed over the paper. An overview section (first paragraph of method section) that summarizes the selection criteria used will better orient the reader.
Acceptance Reason:
Strong results. The problem statement and motivation are clearly stated and illustrated, making it easy to follow the design choices. Substantial novelty in methodology. Explores an important direction for inference-time acceleration of diffusion models.

**Additional Comments On Reviewer Discussion:**

The paper received 3x marginally above acceptance threshold. Reviewers raised a number of points, which were adequately addressed by the authors, and were acknowledged by the reviewers.

---

### Decision · Program_Chairs · 2025-01-22

Accept (Poster)